# MEAL: A Benchmark for Continual Multi-Agent Reinforcement Learning

## Abstract

Benchmarks play a crucial role in the development and analysis of reinforcement learning (RL) algorithms, with environment availability strongly impacting research. One particularly underexplored intersection is continual learning (CL) in cooperative multi-agent settings. To remedy this, we introduce **MEAL** (**M**ulti-agent **E**nvironments for **A**daptive **L**earning), the first benchmark tailored for continual multi-agent reinforcement learning (CMARL). Existing CL benchmarks run environments on the CPU, leading to computational bottlenecks and limiting the length of task sequences. MEAL leverages JAX for GPU acceleration, enabling continual learning across sequences of 100 tasks on a single GPU in a few hours. We show that naïvely combining popular CL and MARL methods yields strong performance on simple environments, but fails to scale to more complex settings requiring sustained coordination and adaptation.

## 1 Introduction

Continual RL has recently attracted growing interest Hafez & Erekmen (2024); Chen et al. (2024); Chung et al. (2024); Erden et al., yet remains largely unexplored in multi-agent settings Yuan et al. (2023; 2024). Combining the two introduces unique challenges. In cooperative environments, agents ought to establish implicit conventions or roles for coordination Strouse et al. (2021). As tasks or dynamics change, these conventions may fail, and inter-agent dependencies can turn individual forgetting into team-wide breakdowns. Unlike traditional MARL, CMARL faces non-stationarity not only from other learning agents but also from evolving task distributions Yuan et al. (2024). This dual source of change demands agents that can adapt and transfer knowledge without discarding prior coordination strategies. This ability is critical in real-world settings where environments continually evolve. For instance, autonomous vehicles must navigate unseen roads, adapt to new traffic regulations, and interact with unfamiliar human drivers, while coordinating with other AVs. Similarly, warehouse robots deployed in a new facility must quickly adapt to unseen layouts and workflows, while preserving established collaborative behaviors.

To analyze how current methods handle the interplay between CL and MARL, and to drive progress in this domain, we introduce **MEAL**[1], the first benchmark for CMARL. To the best of our knowledge, MEAL is also the first continual RL library to leverage JAX for end-to-end GPU acceleration. Traditional CPU-based benchmarks are limited to short sequences (5–15 tasks) due to low environment throughput and task diversity Sorokin & Burtsev (2019); Powers et al. (2022); Tomilin et al. (2023), making them ill-suited for the computational demands of CL across long task sequences. MEAL's end-to-end JAX pipeline removes this barrier, enabling training on up to 100 tasks within a few hours on a single desktop GPU. This unlocks new research directions for scalable, cooperative continual learning in resource-constrained settings.

MEAL is built on Overcooked Carroll et al. (2019), a widely used cooperative MARL environment Hu et al. (2020); Wu et al. (2021); Strouse et al. (2021), providing a strong foundation for benchmarking. Prior work has shown that agents tend to exploit spurious correlations in fixed layouts, resulting in poor generalization even under minor modifications Knott et al. (2021). This makes Overcooked particularly well-suited for learning continually: even minor layout variations can present a significant challenge. To succeed across a sequence of such tasks, agents must avoid overfitting to layout-specific behaviors and instead learn coordination strategies that are robust and transferable.

---

[1]The code and environments are accessible on GitHub.

Table 1: Comparison of existing Reinforcement Learning benchmarks with MEAL.

| Benchmark | PCG | Difficulty Levels | GPU-accelerated | Action Space | Multi-Agent | Continual Learning |
|---|---|---|---|---|---|---|
| CORA | ✓ / ✗ | ✗ | ✓ | Mixed | ✗ | ✓ |
| MPE | ✗ | ✗ | ✗ | Continuous | ✓ | ✗ |
| SMAC | ✗ | ✓ | ✗ | Discrete | ✓ | ✗ |
| Continual World | ✗ | ✗ | ✗ | Continuous | ✗ | ✓ |
| Melting Pot | ✗ | ✗ | ✗ | Discrete | ✓ | ✗ |
| Google Football | ✗ | ✓ | ✓ | Discrete | ✓ | ✗ |
| JaxMARL | ✓ / ✗ | ✗ | ✓ | Mixed | ✓ | ✗ |
| COOM | ✗ | ✓ | ✗ | Discrete | ✗ | ✓ |
| **MEAL** | ✓ | ✓ | ✓ | Discrete | ✓ | ✓ |

The **contributions** of our work are three-fold. (1) We release MEAL, the first CMARL benchmark, consisting of procedurally generated Overcooked environments spanning three difficulty levels. (2) We leverage JAX to build the first end-to-end GPU-accelerated task sequences for continual RL, enabling efficient training on low-budget setups. (3) We implement six popular CL methods in JAX and evaluate them in MEAL, revealing key shortcomings in retaining cooperative behaviors and adapting to shifting roles across tasks.

## 2 RELATED WORK

**Continual Reinforcement Learning (CRL)**  CRL studies how agents can learn sequentially from a stream of tasks without forgetting previous knowledge. A wide range of methods have been adapted from the CL literature to facilitate the RL setting, including regularization-based approaches such as EWC Kirkpatrick et al. (2017), SI Zenke et al. (2017), and MAS Aljundi et al. (2018); architectural strategies such as PackNet Mallya & Lazebnik (2018); and replay-based methods like RePR Atkinson et al. (2021). More recent works focus on scalability Hafez & Erekmen (2024), memory efficiency Chung et al. (2024), and stability during training Chen et al. (2024). However, these methods are almost exclusively developed for single-agent settings, and their behavior under multi-agent coordination remains largely unexplored.

**Multi-Agent Reinforcement Learning (MARL)**  In MARL, multiple agents learn to act in a shared environment, often with partial observability and either cooperative or competitive goals Hernandez-Leal et al. (2019); OroojlooyJadid & Hajinezhad (2019). A major focus has been on cooperative settings, where agents share a reward function and must learn to coordinate Lowe et al. (2017); Foerster et al. (2018). Popular algorithms include IPPO De Witt et al. (2020), VDN Sunehag et al. (2017), QMIX Rashid et al. (2020), and MAPPO Yu et al. (2022).

**Continual MARL**  Research on CMARL is sparse. Lifelong Hanabi Nekoei et al. (2021) introduces a testbed for evaluating whether agents can coordinate with unseen teammates. The MACPro framework Yuan et al. (2024) proposes a continual coordination approach, using learned task contextualization and progressive multi-head expansion to handle evolving tasks.

**Benchmarks**  Standard CRL benchmarks include Continual World Wołczyk et al. (2021), COOM Tomilin et al. (2023), and CORA Powers et al. (2022). While effective in single-agent settings, they either lack multi-agent capabilities or suffer from slow CPU-bound environments. SMAC Samvelyan et al. (2019), MPE Mordatch & Abbeel (2018), Google Football Kurach et al. (2020), and Melting Pot Agapiou et al. (2022) are widely used for MARL, but are not designed for continual learning. Overcooked Carroll et al. (2019) has emerged as a useful domain for studying coordination, with recent implementations in JAX Rutherford et al. (2024b). Our benchmark builds on Overcooked and introduces procedural variation to create long task sequences for continual MARL.

**Overcooked**  The Overcooked environment Carroll et al. (2019) is a cooperative multi-agent benchmark inspired by a popular video game, where high performance requires strategic collaborative behaviors. Agents control chefs in a grid-based kitchen to prepare and deliver dishes through

sequences of interactions with pots, ingredient dispensers, plate stations, and delivery counters. Compared to the large state spaces and high agent counts in benchmarks like Melting Pot and SMAC, Overcooked operates on small grid-based environments with few agents. However, its complexity arises not from scale but from credit assignment challenges, and the need for precise coordination, as agents must execute tightly coupled action sequences (Hernandez-Leal et al., 2019).

## 3 PRELIMINARIES

**Cooperative Multi-Agent MDP**  We formulate the setting as a fully observable cooperative multi-agent task, modeled as a Markov game defined by the tuple $\langle N, S, A, P, R, \gamma \rangle$, where $N$ is the number of agents, $S$ is the state space, $A^i$ is the action space of agent $i$ with joint action space $A = A^1 \times \cdots \times A^N$, $P : S \times A \times S \to [0, 1]$ is the transition function, $R : S \times A \times S \to \mathbb{R}$ is a shared reward function, and $\gamma \in [0, 1)$ is the discount factor. In the fully observable setting, each agent receives the full state $s \in S$ at every time step.

**Continual MARL**  We consider a continual MARL setting in which a shared policy $\pi_\theta = \pi_{\theta i \in N}^i$ is learned over a sequence of tasks $\mathcal{T} = \mathcal{M}_1, \ldots, \mathcal{M}_N$, where each $\mathcal{M}_i = \langle N, S_i, A, P_i, R_i, \gamma \rangle$ is a fully observable cooperative Markov game with consistent action and observation spaces. At training phase $i$, agents interact exclusively with $\mathcal{M}_i$ for a fixed number of iterations $\Delta$, collecting trajectories $\tau_{i,1}, \ldots, \tau_{i,\Delta}$ to update their policy. Past tasks are inaccessible, and no joint training is allowed. The objective is to maximize performance on all tasks in the sequence.

## 4 MEAL

We present MEAL, the first CMARL benchmark, built on the JaxMARL Rutherford et al. (2024b) version of Overcooked. The goal in Overcooked is for agents to cooperatively prepare and deliver soup. They must collect onions, place them into pots, wait for the soup to cook, plate the dish, and deliver it to a serving station. We leverage JAX Bradbury et al. (2018) for efficient environment simulation. JAX provides just-in-time compilation, automatic differentiation, and vectorization through XLA, enabling high-performance computation. In the scope of this work, we deliberately focus solely on Overcooked to allow for deeper analysis rather than spread thinly across many domains. We go beyond reporting standard CL performance metrics and analyze the core challenges unique to cooperative CMARL, including coordination under changing layouts, shifts in roles and division of labour, and adaptation to evolving partner behaviour. To demonstrate MEAL's extensibility, we incorporate JAXNAV Rutherford et al. (2024a) (Appendix K).

### 4.1 ENVIRONMENT SPECIFICATIONS

**Dynamics**  Agents act synchronously at each time step. Moves into walls or occupied tiles are no-ops, and simultaneous swaps are disallowed (both agents remain in place). Agents can `interact` with the tile they are facing, which deterministically updates the object's state (pick/place, add onion, plate, deliver). Pots initiate a fixed cook timer of $c_{\text{cook}} = 20$ steps when the third onion is added, and the cooked soup can only be plated upon completion.

**Observations**  Each agent receives a fully observable grid-based observation of shape $(H, W, 26)$, where $H$ and $W$ are the height and width of the grid, and the 26 channels encode tile types (e.g., walls, agents, onions, plates, pots, delivery stations) and states (e.g., cooking progress, held item). To maintain a consistent observation space for CL, we fix the shape to $(H_{\text{max}}, W_{\text{max}}, 26)$, where $H_{\text{max}}$ and $W_{\text{max}}$ are the largest grid dimensions in the sequence, and pad all smaller layouts with walls.

**Action Space**  At each timestep, both agents select one of six discrete actions from a shared action space $\mathcal{A} = \{\text{up, down, left, right, stay, interact}\}$. Movement actions translate the agent forward if the target tile is free (i.e., not a wall or occupied), while `stay` maintains the current position. The `interact` action is context-dependent and allows agents to pick up or place items, add ingredients to pots, serve completed dishes, or deliver them at the goal location. Importantly, there is no built-in communication action; all coordination emerges from environment interactions.

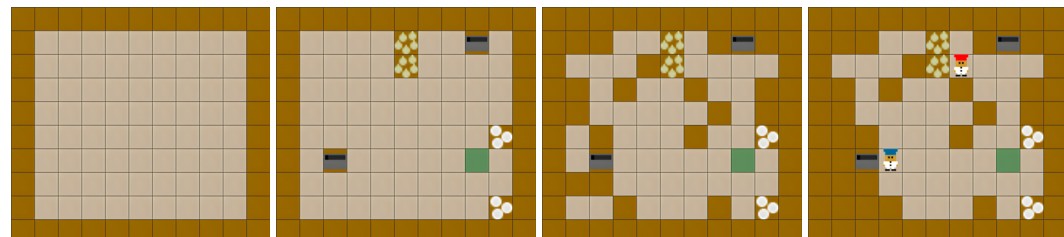

(a) Empty grid drawn with outer walls.
(b) Interactive stations sampled at random locations.
(c) Grid filled with walls to match obstacle density.
(d) Agents added and unreachable tiles pruned.

Figure 1: Procedural generation pipeline of a **hard** layout. Starting from an empty grid with outer walls, the generator injects interactive stations, adds walls to match the desired obstacle density, places agents, and finally prunes unreachable tiles.

**Rewards** Agents receive a shared team reward: $r_t = r_{\text{deliver}} + r_{\text{onion}} \cdot \mathbb{1}_{\{\text{onion\_in\_pot}\}} + r_{\text{plate}} \cdot \mathbb{1}_{\{\text{plate\_pickup}\}} + r_{\text{soup}} \cdot \mathbb{1}_{\{\text{soup\_pickup}\}}$, where $r_{\text{deliver}} = 20$ is the reward for delivering soup, and the other terms provide shaped rewards for intermediate progress. We include two reward settings: in the **sparse** setting, $r_{\text{onion}} = r_{\text{plate}} = r_{\text{soup}} = 0$; in the **dense** setting, $r_{\text{onion}} = r_{\text{plate}} = 3$, and $r_{\text{soup}} = 5$.

## 4.2 MEAL GENERATOR

Existing continual RL benchmarks only provide a fixed set of tasks Sorokin & Burtsev (2019); Powers et al. (2022); Tomilin et al. (2023). To avoid over-fitting to a fixed set of environments, we procedurally generate new Overcooked kitchens on the fly. The generator $G$ draws a random width and height from the specified range, places an outer wall, then sequentially injects the interactive tiles (goal, pot, onion pile, plate pile), extra internal walls to match the target obstacle density, and finally, the agents' starting positions. Figure 1 depicts the steps in the pipeline, and the process is described more in-depth in Appendix A.2. Each candidate grid is accepted only if a built-in validation module confirms that both agents can complete at least one cook–deliver cycle. This yields a continuous space of solvable, variable-sized kitchens that we can learn continually. We bring further details about the validator in Appendix A.3. Our approach offers a virtually infinite supply of tasks and evaluates true lifelong learning under continual exposure to unseen configurations. To ensure reproducibility and a fair comparison between methods, the generation process can be fully controlled via a user-specified random seed.

## 4.3 LAYOUT DIFFICULTY

We categorize environment difficulty based on procedurally generated layout characteristics. We vary the (1) grid width, (2) grid height, and (3) obstacle density. This approach produces diverse spatial configurations while maintaining consistent difficulty within each level. Figure 2 depicts layouts of each difficulty. As grid size and the number of impassable tiles increase, agents must develop more sophisticated coordination strategies. Higher difficulty layouts feature longer paths between key items, tighter bottlenecks, and greater structural variability, all of which make exploration, retention, and adaptation more challenging. Level 1 tasks are designed to be simple enough for existing methods to achieve reasonably high scores, enabling better comparisons and behavioral analysis. Higher levels are intended to challenge future methods. Although we currently include three difficulty levels, it is straightforward to extend the framework. Although obstacle density has a practical upper bound, the grid size can be increased arbitrarily to scale up environment complexity.

## 4.4 CONTINUAL LEARNING SEQUENCES

**Kitchen Layouts** MEAL provides discrete task sequences $\mathcal{T} = (\mathcal{M}_1, \ldots, \mathcal{M}_N)$ rather than a continuous domain shift. For a chosen difficulty level $\ell \in \{1, 2, 3\}$, we sample $N$ solvable layouts i.i.d. from the generator $G_\ell$ with a fixed seed. At task boundaries, we carry over the optimizer state and policy parameters, reset rollout buffers, and advance the RNG. We explore three sequence regimes: (i) **fixed-level**, where all tasks are drawn from the same difficulty level; (ii) **curriculum**, where

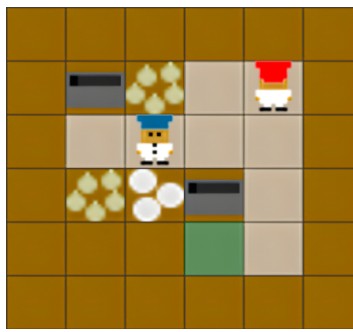 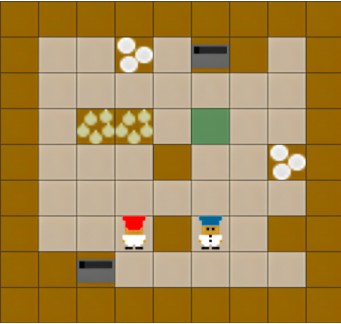 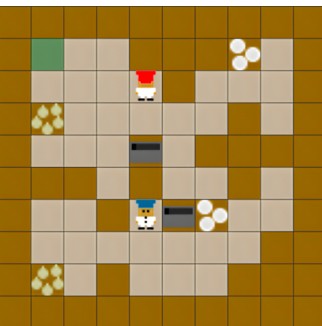

(a) **Level 1** (Easy): $6 \leq$ width/height $\leq 7$, obstacle density $\approx 15\%$. Layouts are compact, making exploration easy. Interactable items are close together, making travel distances short. Agents can often complete the task independently with no coordination.

(b) **Level 2** (Medium): $8 \leq$ width/height $\leq 9$, obstacle density $\approx 25\%$. Exploration is harder as stations are more spread out. Layouts often introduce chokepoints, requiring agents to coordinate movement and avoid congestion.

(c) **Level 3** (Hard): $10 \leq$ width/height $\leq 11$, obstacle density $\approx 35\%$. Layouts are likely to split the map into disjoint regions, forcing agents to specialize. Solving the task requires deliberate cooperation and division of labor.

Figure 2: Overcooked layouts generated at each difficulty level. Increasing grid size and obstacle density lead to longer travel distances, harder exploration, and greater coordination demands.

sequences contain an equal number of tasks in increasing difficulty level (see Appendix E), and (iii) **repetition**, where a sequence is repeated $r$ times to study the loss of network plasticity (section 5.5).

**Diverse Partners**   Ad-Hoc Teamwork (Stone et al., 2010, AHT) is the task of coordinating with unknown partners. AHT algorithms are typically evaluated with diverse partner populations as a proxy for human-AI coordination performance and to test robustness to a diverse set of strategies Yan et al. (2023); Wang et al. (2025); Ruhdorfer et al. (2025b). Following prior work Wang et al. (2025); Ruhdorfer et al. (2025a), we generate diverse evaluation partners by combining (i) hardcoded strategies (random, static), (ii) planning-based agents (onion-only, plate-only, and a human-like planner with stochastic task selection), and (iii) populations trained with best-response diversity (Rahman et al., 2023, BRDiv), which maximizes self-play performance while minimizing cross-play compatibility. Opposed to much prior work on AHT that targets zero-shot human–AI coordination Strouse et al. (2021); Zhao et al. (2023); Yan et al. (2023), MEAL provides the tools to test how agents can continually learn to adapt to novel partners. We evaluate this setting in Appendix J.

### 4.5   EVALUATION METRICS

We measure task performance by the number of soups delivered per episode. Since MEAL layouts vary greatly in size, structure, number of interactive stations, and distances between them, raw delivery counts are not directly comparable. We therefore normalize the delivery count by the optimal cook-deliver cycle for a single agent on any given task (see Appendix A.1). We account for the cooking time, pickup/drop interactions, shortest paths between onion piles, pots, plate piles, and delivery counters. A score of 1 indicates that the agent(s) achieved the optimal single-agent performance, while values above 1 reflect effective cooperation that exceeds solo efficiency. Let $s_i(t)$ denote this normalized delivery score on task $i$ at timestep $t$. Suppose that the training sequence consists of $N$ tasks, each lasting $\Delta$ steps, resulting in a total of $T = N \cdot \Delta$ timesteps. The $i$-th task is therefore trained during the interval $t \in [(i-1)\Delta, i\Delta]$. Following prior work on CL Wołczyk et al. (2021); Tomilin et al. (2023), we rely on three metrics.

**Average Normalized Score**   To capture the balance between stability and plasticity, we report the mean score across all tasks at the end of training:

$$\mathcal{A} = \frac{1}{N} \sum_{i=1}^{N} s_i(T). \tag{1}$$

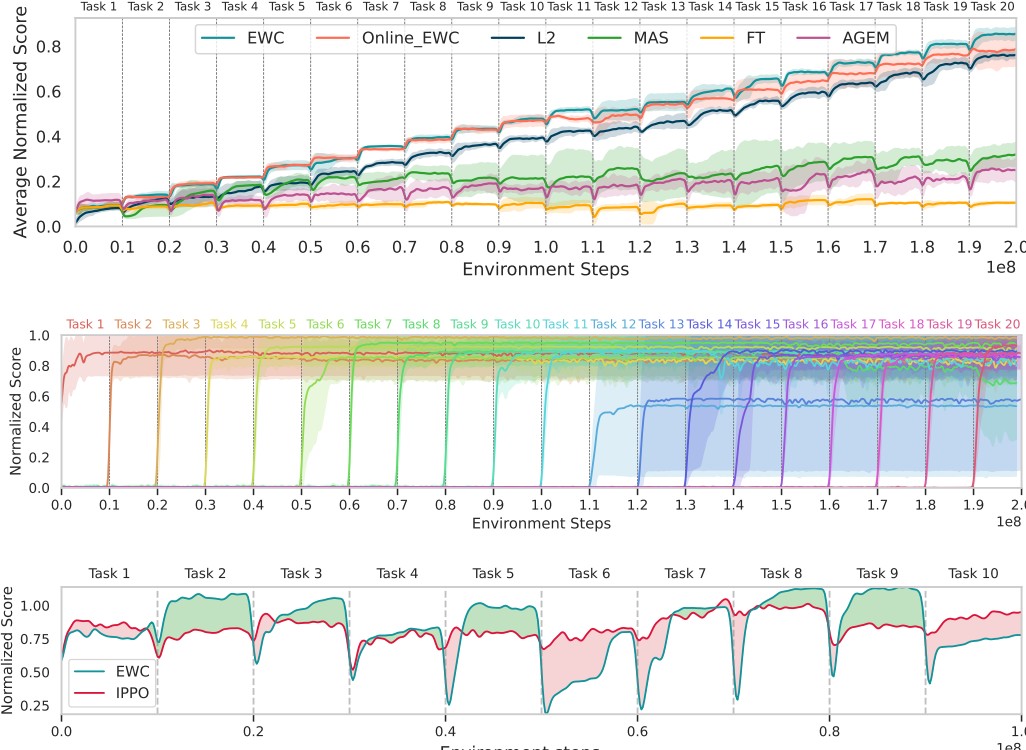

Figure 3: **Top:** The average normalized score evaluation curves on Level 1 tasks show a notable performance gap across baselines. **Middle**: Per-task evaluation scores for EWC on Level 1 indicate near-perfect retention. **Bottom**: EWC manages to outperform the standard IPPO baseline by transferring knowledge forward on most Level 2 tasks, while under-performing in others. The green area between the curves indicates positive forward transfer, while red represents the negative counterpart.

**Forgetting**   Forgetting quantifies the decline in performance on past tasks due to interference from training on later ones. For each task $i < N$, let $\tau_i = i \cdot \Delta$ be the timestep when its training finishes and $s_i^\star = s_i(\tau_i)$ the score at that moment. We define the normalized drop $d_i(t)$ for $t > \tau_i$ and assign exponentially decaying weights $w_i(t)$ (with decay factor $\lambda > 0$) to penalize earlier forgetting more strongly. The overall forgetting is the average across tasks:

$$d_i(t) \;=\; \max\Big(0, \tfrac{s_i^\star - s_i(t)}{s_i^\star}\Big), \quad w_i(t) \;=\; e^{-\lambda \frac{t-\tau_i}{T-\tau_i}}, \quad \mathcal{F} \;=\; \frac{1}{N-1} \sum_{i=1}^{N-1} \frac{\sum_{t>\tau_i} w_i(t)\, d_i(t)}{\sum_{t>\tau_i} w_i(t)}. \quad (2)$$

**Forward Transfer**   Forward transfer measures how prior experience accelerates the learning of new tasks. Rather than evaluating final performance, it captures how quickly each task is learned relative to a single-task baseline. We compute the normalized area under the learning curves (AUC) for both the CL agent and the baseline. The area difference between these curves is positive when prior training helps, and negative when it hinders.

$$\mathrm{AUC}_i = \frac{1}{\Delta} \int_{(i-1)\Delta}^{i\Delta} s_i(t)\, dt, \quad \mathrm{AUC}_i^{\mathrm{b}} = \frac{1}{\Delta} \int_0^{\Delta} s_i^{\mathrm{b}}(t)\, dt. \quad \mathcal{FT} = \frac{1}{N} \sum_{i=1}^{N} \frac{\mathrm{AUC}_i - \mathrm{AUC}_i^{\mathrm{b}}}{1 - \mathrm{AUC}_i^{\mathrm{b}}}. \quad (3)$$

## 5   EXPERIMENTS

The agent is trained on each task $\mathcal{T}_i$ for $\Delta = 10^7$ environment steps on-policy with the dense reward setting, repeated over five seeds. In our experiments, we adopt the **task-incremental** continual learning paradigm, in which the task identity is known during both training and evaluation. During training, we evaluate the policy after every 100 updates by running 10 evaluation episodes on all

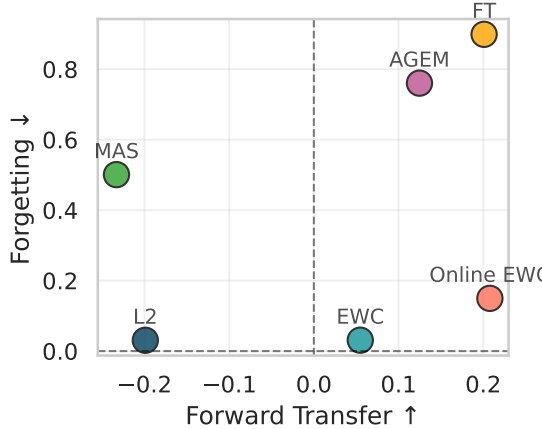
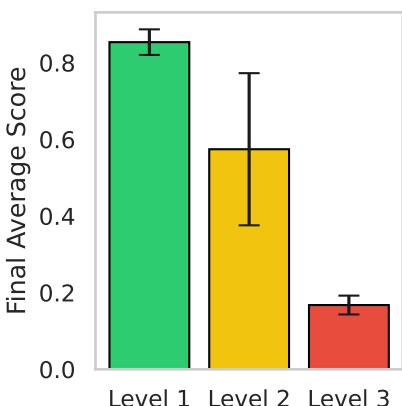

Figure 4: Jointly visualizing forward transfer and forgetting results on Level 1 reveals the classic stability–plasticity trade-off in continual learning.

Figure 5: EWC's performance notably declines as layout complexity increases. Most high-level tasks remain unsolved.

Table 2: Baseline comparison results across three difficulty levels. The confidence intervals are omitted for brevity; see Appendix M for the full results.

| Method | Level 1 | | | Level 2 | | | Level 3 | | |
|---|---|---|---|---|---|---|---|---|---|
| | $\mathcal{A}\uparrow$ | $\mathcal{F}\downarrow$ | $\mathcal{FT}\uparrow$ | $\mathcal{A}\uparrow$ | $\mathcal{F}\downarrow$ | $\mathcal{FT}\uparrow$ | $\mathcal{A}\uparrow$ | $\mathcal{F}\downarrow$ | $\mathcal{FT}\uparrow$ |
| FT | 0.048 | 0.946 | 0.201 | 0.041 | 0.944 | 0.065 | 0.010 | 0.903 | -0.157 |
| EWC | **0.839** | **0.012** | 0.055 | **0.604** | **0.026** | -0.086 | 0.178 | 0.082 | -0.650 |
| Online EWC | 0.769 | 0.062 | **0.208** | 0.585 | 0.096 | **0.152** | **0.306** | 0.141 | **-0.149** |
| MAS | 0.281 | 0.286 | -0.233 | 0.155 | 0.309 | -0.355 | 0.034 | 0.380 | -0.542 |
| L2 | 0.753 | 0.018 | -0.199 | 0.496 | 0.058 | -0.527 | 0.127 | **0.070** | -0.827 |
| AGEM | 0.204 | 0.678 | 0.125 | 0.117 | 0.801 | -0.083 | 0.037 | 0.861 | -0.169 |

tasks in the sequence. The results are displayed with 95% confidence intervals. We leverage JAX to reduce the wall-clock time for training on a single task to around 5 minutes. Our experiments are conducted on a dedicated compute node with a 72-core 3.2 GHz AMD EPYC 7F72 CPU and a single NVIDIA A100 GPU. We adopt many of JaxMARL's default settings for our network configuration, IPPO setup, and training processes. For exact hyperparameters please refer to Appendix B.2.

## 5.1 BASELINE COMPARISON

We evaluate popular CL methods. Fine-Tuning (**FT**) is a naive baseline where the policy is trained sequentially across tasks without any mechanism to prevent forgetting. **L2-Regularization** Kirkpatrick et al. (2017) adds a penalty on parameter changes to promote stability. **EWC** (Elastic Weight Consolidation) Kirkpatrick et al. (2017) penalizes changes to important parameters, with importance measured using the Fisher Information Matrix. **Online EWC** is a variant that maintains a running estimate of parameter importance. **MAS** (Memory Aware Synapses) Aljundi et al. (2018) computes importance based on how parameters influence the policy's output, rather than gradients. **AGEM** (Averaged Gradient Episodic Memory) Chaudhry et al. (2018) is a replay-based method that projects the current gradient update to avoid interference with past tasks, using a memory buffer of stored experiences. As the default MARL algorithm, we opt for **IPPO** De Witt et al. (2020). It is a natural choice as it can be seamlessly integrated with all model-free CL methods. Moreover, it has been shown to outperform other MARL approaches on SMAC De Witt et al. (2020) and Overcooked Rutherford et al. (2024b), making it a strong candidate for evaluating CMARL in MEAL.

Figure 3 (top) compares our baselines on Level 1, and Table 2 reports the exact metrics for all levels. Fine-Tuning (FT) and AGEM show high $\mathcal{FT}$, but exhibit immediate forgetting once a task is left behind. EWC and L2 show near-perfect retention on all levels, with EWC ranking highest in $\mathcal{A}$ on lower levels. Figure 3 (middle) visualizes EWC's per-task stability. Refer to Appendix I.1 for a deeper

Table 3: Comparison of EWC with PPO/IPPO across 1–3 agent task sequences. Two agents yield the best results due to parallelism and cooperative potential. Adding a third agent introduces instability, non-stationarity, and coordination challenges that hurt performance.

| Agents | Level 1 | | | Level 2 | | | Level 3 | | |
|---|---|---|---|---|---|---|---|---|---|
| | $\mathcal{A}\uparrow$ | $\mathcal{F}\downarrow$ | $\mathcal{FT}\uparrow$ | $\mathcal{A}\uparrow$ | $\mathcal{F}\downarrow$ | $\mathcal{FT}\uparrow$ | $\mathcal{A}\uparrow$ | $\mathcal{F}\downarrow$ | $\mathcal{FT}\uparrow$ |
| 1 Agent | 0.622 | 0.046 | -0.045 | 0.343 | 0.071 | -0.458 | 0.285 | 0.159 | **-0.531** |
| 2 Agents | 0.839 | **0.031** | **0.055** | 0.604 | **0.061** | **-0.086** | 0.178 | **0.053** | -0.650 |
| 3 Agents | **0.860** | 0.197 | -0.129 | **0.647** | 0.212 | -0.295 | **0.303** | 0.211 | -0.613 |
| 4 Agents | 0.361 | 0.143 | -0.714 | 0.336 | 0.223 | -0.574 | 0.118 | 0.109 | -0.901 |

analysis of EWC. Per-task evaluation curves of other baselines can be found in Figure 21. These curves reveal the full forgetting dynamics across tasks: FT and AGEM collapse immediately after each switch, MAS drifts steadily, and L2 maintains a stable performance throughout the sequence. MAS performs poorly in all metrics, although outperforming FT and AGEM. Notably, the simplistic difficulty level design of MEAL presents notable challenges, as EWC's score diminishes with increasing difficulty (Figure 5). Figure 4 illustrates the fundamental stability-plasticity trade-off in CL. L2 achieves excellent retention but limited forward transfer, while Fine-Tuning and AGEM demonstrate high plasticity with severe forgetting. EWC and its online version provide a middle ground, balancing both objectives more effectively than other approaches.

## 5.2 Ablation Study

To determine which components are crucial for CMARL on MEAL, we ablate five components in our default IPPO learning setup: multi-head architectures, task identity inputs, critic regularization, layer normalization, and replacing the MLP with a CNN encoder. The results in Figure 6 reveal that multi-head outputs are most critical for MEAL task sequences. Removing them consistently devastates performance across all methods, likely due to uncontrolled interference between tasks in the shared output head. In contrast, not provid-

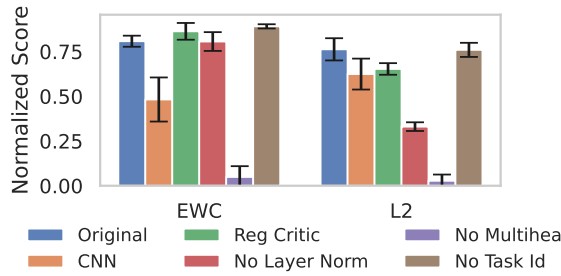

Figure 6: **Ablation results on Level-1.** The multi-head architecture is the most beneficial component, while the task ID and critic regularization have negligible effect. Layer normalization improves L2.

ing the model with the one-hot encoded task ID vector has a negligible effect. Prior continual RL studies Wołczyk et al. (2021); Tomilin et al. (2023) report that it is beneficial to only regularize the actor and let the critic adapt freely. In our experiments, however, we find that this has little effect. Layer normalization shows method-specific sensitivity: while it makes little difference for EWC and MAS, it more than doubles the performance of L2 regularization. This is likely because L2 penalizes absolute weight magnitudes, and layer norm helps stabilize activations across tasks, mitigating harmful scale drift. Finally, swapping to a CNN encoder substantially hurts performance for all methods. Given the small layouts in Level 1 tasks (6×6 to 7×7), CNNs struggle to extract meaningful features and add unnecessary parameter overhead, making simple MLPs the better fit in this setting.

## 5.3 $N$-Agent MEAL

To better analyze the multi-agent dimension of CMARL, we extend MEAL to support an $N$-agent setting, allowing us to systematically study how the number of cooperating agents affects CL. We run EWC combined with PPO for a single agent, and IPPO for multiple agents. The single agent delivers fewer soups because it cannot parallelize tasks (Table 3): while one agent delivers soup, the other can already refill the pot with onions. However, in Level 3, the extra agents and larger grid size increase the observation space leading to worse performance. IPPO trains independent policies while the environment remains a joint MDP, where transitions and rewards depend on the combined actions of all agents. Moving from 2→3 agents expands the joint action space and interaction patterns, amplifies

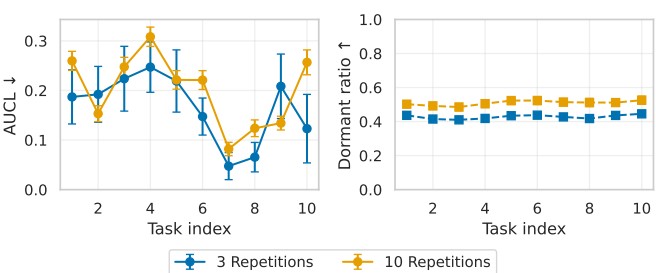

Table 4: Averaged plasticity metrics on Level-1 task sequences. The larger drop in performance occurs between 1 and 3 repetitions, suggesting that early degradation is more severe. AUC-loss increases by roughly 40% when going from 3→10 repetitions.

Figure 8: **Loss of plasticity in MEAL.** AUCL (**left**) captures performance loss, and the Dormancy ratio (**right**) quantifies the fraction of inactive neurons. Increasing the repetition count leads to lower performance and more dormant neurons.

| Reps | AUCL ↓ | Dormancy ↓ |
|------|--------|------------|
| 1    | 0.000  | 0.408      |
| 3    | 0.166  | 0.428      |
| 10   | 0.201  | 0.509      |

non-stationarity (as two teammates' policies change simultaneously), and makes credit assignment more difficult (since the reward is shared, IPPO does not know which agent made a good action). Without explicit communication or role allocation, IPPO struggles to learn continually as the team and layout size grow. These results indicate that CL becomes increasingly challenging as the number of agents grows. We explore common pitfalls of agent behavior in Appendix H.

## 5.4 PARTIAL OBSERVABILITY

Although Overcooked is fully observable by design, we introduce a partially observable variant to better reflect real-world sensing constraints (limited field of view, occlusions). Following popular MARL environments Resnick et al. (2018); Mohanty et al. (2020); Agapiou et al. (2022); Ellis et al. (2023), each agent receives an egocentric, direction-aware observation window with all outside tiles masked. The specification and difficulty scaling of this window are detailed in Appendix C. In this setting, MAPPO Yu et al. (2022) is known to outperform IPPO, since its centralized critic can 1) more accurately estimate individual contributions to shared rewards under partial observability, and 2) reduce non-stationarity by conditioning value estimates on the joint actions of all agents, leading to more stable and coordinated policy updates. We investigate this by running a 20-task sequence under partial observability (PO) with EWC and compare the results with the fully observable (FO) baseline.

Across all levels, IPPO (FO) clearly dominates the partial setting (PO) (Figure 7). The gap between IPPO(FO) and IPPO(PO) stems from full state information simplifying credit assignment and stabilizing value targets. Partial observability thus increases task difficulty. Contrary to expectation, MAPPO underperforms IPPO. A plausible cause is a mismatch between MAPPO's centralized critic and the CL regime. Conditioning on joint observations and actions drifts substantially across tasks, yielding noisier targets and stronger cross-task interference, while IPPO's independent critics learn simpler task-local value functions that transfer more stably. These results motivate including PO variants in MEAL to stress coordination under incomplete information for more realistic continual MARL benchmarking.

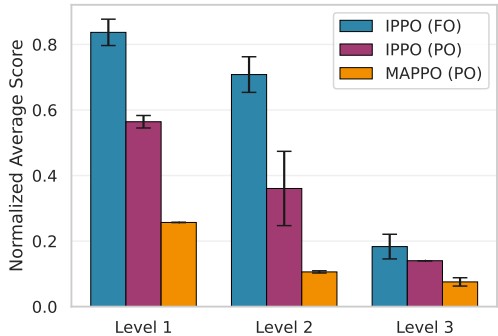

Figure 7: Performance of EWC under full (FO) and partial (PO) observability on 20-task Level 1 sequences. FO yields better results, while MAPPO underperforms in the CL setting.

## 5.5 NETWORK PLASTICITY

A well-documented pitfall in continual RL is the gradual loss of **plasticity**, an agent's ability to fit new data after many tasks (Abbas et al., 2023; Dohare et al., 2024). To test whether MEAL exhibits the

same pathology, we continually train IPPO on a Level 1 10-task sequence across multiple repetitions and compare performance between them. We track two metrics: (i) **AUC-loss** captures capacity drop, (ii) **Dormancy ratio** quantifies the fraction of inactive neurons in the policy network. For definitions of metrics and additional training curves, see Appendix F. We observe that all metrics deteriorate with longer training (Table 4 and Figure 8), confirming that loss of plasticity also appears in the multi-agent setting. Despite our setting spanning over 1B environment steps, well beyond the scale of prior studies (Abbas et al., 2023; Dohare et al., 2024), those works report a much stronger loss of plasticity than observed in MEAL. We hypothesize that this difference stems from our experiments using multiple output heads, which isolate task-specific outputs, reduce gradient interference, and preserve prior policies while allowing the backbone to learn transferable features.

## 6 LIMITATIONS

While MEAL provides a scalable and diverse testbed for CMARL, several limitations remain. First, MEAL is restricted to discrete action spaces, limiting its applicability. Second, while layout diversity is high, the domain itself is narrow. Overcooked dynamics do not capture the full complexity of real-world multi-agent interactions involving language, negotiation, or long-horizon planning. Third, our benchmark only evaluates task-incremental learning by changing layouts, partners, and factors of non-stationarity. Future work could extend MEAL to other CL settings. Finally, although we intentionally focused on a single Overcooked domain, other domains and settings remain largely unexplored.

## 7 CONCLUSION

We introduced MEAL, a scalable benchmark for CMARL, built on JAX for efficient GPU training. The on-demand creation of procedurally generated Overcooked layouts enables long-horizon studies with controlled difficulty, observability, and agents. We evaluated combinations of popular CL methods and MARL algorithms, revealing that existing techniques struggle to retain cooperative behaviors while maintaining adaptability to new tasks. The $N$-agent setting increases coordination demands and interference, yielding a harder, more variable task distribution. Partial observability compounds this difficulty, as centralized critics exhibit stronger cross-task drift and interference. Individual rewards weaken coordination and induce negative transfer. A simple curriculum boosts performance on complex layouts under an equal data budget. Training on long task sequences degrades network plasticity in MARL, while multi-head architectures yield the largest structural gains for performance. Our findings suggest that MEAL exposes the dual challenge of cooperation and non-stationarity in CMARL. We see immediate headroom for methods that (i) are purpose-built for CMARL, jointly handling partner and environment-level non-stationarity, (ii) stabilize credit assignment under partial observability across task sequences, and (iii) drive structured exploration and robust coordination in diverse, long-horizon settings. We hope MEAL serves as a solid foundation for pushing this line of work forward.

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

# A  IMPLEMENTATION DETAILS

## A.1  MAXIMUM SOUP DELIVERY CALCULATOR

Let a kitchen layout $\mathcal{L}$ be defined by four disjoint sets of tiles (onion piles $\mathcal{O}$, plate piles $\mathcal{P}$, pots $\mathcal{K}$, delivery counters $\mathcal{G}$) and a set of walls $\mathcal{W}$. A tile $(x, y)$ is *walkable* if $(x, y) \notin \mathcal{W}$.

**Neighbourhood of an object family.**  We denote the set of walkable tiles adjacent (in the 4-neighbour sense) to any object in $\mathcal{S}$ as:

$$\mathcal{N}(\mathcal{S}) \;=\; \big\{(x', y') \mid (x, y) \in \mathcal{S},\; \|(x', y') - (x, y)\|_1 = 1,\; (x', y') \notin \mathcal{W}\big\}$$

**Shortest obstacle-aware distance.**  Given two tile sets $A, B \subseteq \mathbb{Z}^2$, we define

$$d(A, B) \;=\; \min_{a \in A,\, b \in B}\; \mathrm{dist}^{\mathcal{G}_\mathcal{L}}_{\mathrm{manhattan}}(a, b),$$

where $\mathcal{G}_\mathcal{L}$ is the grid graph induced by walkable tiles. We realize this via a breadth-first search (BFS).

**Single-agent cook–deliver cycle.**  A soup requires three onions, one plate pick-up, one soup pick-up, and one delivery. Let

$$d_{\mathrm{onion}} = d\big(\mathcal{N}(\mathcal{O}), \mathcal{N}(\mathcal{K})\big), \quad d_{\mathrm{plate}} = d\big(\mathcal{N}(\mathcal{P}), \mathcal{N}(\mathcal{K})\big), \quad d_{\mathrm{goal}} = d\big(\mathcal{N}(\mathcal{K}), \mathcal{N}(\mathcal{G})\big).$$

The optimistic *movement* cost for one cycle is

$$c_{\mathrm{move}} \;=\; 3\, d_{\mathrm{onion}} + d_{\mathrm{plate}} + 1 + d_{\mathrm{goal}} + 3.$$

**Interaction overhead.**  Every pick-up or drop is assumed to take a constant $c_{\mathrm{act}} = 2$ steps (turn + interact). With $n_{\mathrm{int}} = 3{\times}2 + 1 + 1 + 1 = 9$ interactions per cycle, the overhead is $c_{\mathrm{over}} = n_{\mathrm{int}}\, c_{\mathrm{act}} = 18$.

**Cycle time and upper bound.**  Including the fixed cooking time $c_{\mathrm{cook}} = 20$ steps, the single-agent cycle time is

$$T_{\mathrm{cycle}} \;=\; c_{\mathrm{move}} + c_{\mathrm{cook}} + c_{\mathrm{over}}.$$

For an episode horizon $H$, we upper-bound the number of soups by

$$N_{\mathrm{max}}(\mathcal{L}, H) \;=\; \big\lfloor H / T_{\mathrm{cycle}} \big\rfloor,$$

and convert it to reward with $r_{\mathrm{deliver}} = 20$:

$$R_{\mathrm{max}}(\mathcal{L}, H) \;=\; 20\, N_{\mathrm{max}}(\mathcal{L}, H).$$

The bound assumes *a single agent acting optimally*. It ignores multi–agent collaboration and therefore *underestimates* throughput in layouts where multiple agents can parallelize the workflow. Listing 1 contains the exact implementation.

## A.2  PROCEDURAL KITCHEN GENERATOR

**Objective.**  Given a random seed and user-selectable parameters (number of agents $n_a$, layout height range $[h_{\mathrm{min}}, h_{\mathrm{max}}]$, layout width range $[w_{\mathrm{min}}, w_{\mathrm{max}}]$, and wall-density $\rho$), the goal is to emit a *solvable* grid string $G$ representing the `Overcooked` environment.

### A.2.1  NOTATION

Let $h, w \sim \mathrm{UniformInt}(h_{\mathrm{min}}, h_{\mathrm{max}})$, $\mathrm{UniformInt}(w_{\mathrm{min}}, w_{\mathrm{max}})$, and denote by $\mathcal{C} = \{(i, j) \mid 1 \leq i \leq h - 2,\; 1 \leq j \leq w - 2\}$ the set of *internal* cells (outer walls excluded). Its cardinality is $N_{\mathrm{int}} = (h - 2)(w - 2)$. An *unpassable* cell contains either a hard wall (#) or an interactive tile; we write $N_{\mathrm{unpass}}(G)$ for the number of such cells in $G$.

---

**Listing 1** Heuristic upper bound (`calculate_max_soup`).

---

```python
# overcooked_upper_bound.py      (excerpt)
COOK_TIME = 20
ACTION_OVERHEAD = 2
INTERACTIONS_PER_CYCLE = 3 * 2 + 1 + 1 + 1
OVERHEAD_PER_CYCLE = INTERACTIONS_PER_CYCLE * ACTION_OVERHEAD

def calculate_cycle_time(layout, n_agents=2):
    ...
    move_cost = 3 * d_onion + d_plate + 1 + d_goal + 3
    return move_cost + COOK_TIME + OVERHEAD_PER_CYCLE

def calculate_max_soup(layout, episode_len, n_agents=2):
    cyc = calculate_cycle_time(layout, n_agents)
    soups = episode_len // cyc
    return int(soups)
```

---

### A.2.2  ALGORITHM

The generator performs the following loop until a valid grid is produced (Listing 2):

1. **Draw size.** Sample $h, w$ and create an $h \times w$ matrix initialised to FLOOR tiles, then overwrite the border with WALL.

2. **Place interactive tiles.** For each symbol in {GOAL, POT, ONION_PILE, PLATE_PILE} choose a random multiplicity $m \in \{1, 2\}$ and stamp the symbol onto $m$ uniformly chosen floor cells.

3. **Inject extra walls.** Let $n_{\text{target}} = \lceil \rho\, N_{\text{int}} \rceil$ and $n_{\text{add}} = \max\big(0,\, n_{\text{target}} - N_{\text{unpass}}(G)\big)$. Place $n_{\text{add}}$ additional walls on random floor cells.

4. **Place agents.** Stamp $n_a$ AGENT symbols on random remaining floor cells.

5. **Validate.** Run the deterministic `evaluate_grid` solver; if it returns `True`, terminate and return $(G)$, otherwise restart.

6. **Cleanup.** Remove any interactive elements and tiles that are unreachable from all agent positions.

7. **Return.** Output the final grid.

**Solvability criterion.** The validator (Appendix A.3) checks (i) path connectivity between every agent and each interactive tile family, (ii) at least one pot reachable from an onion pile and a plate pile, and (iii) at least one goal reachable from a pot. This is implemented via multiple breadth-first searches. Appendix A.3 further details the evaluator logic.

**Wall-density effect.** Because interactive tiles themselves count as obstacles, the algorithm first places them, then *only as many extra walls as needed* to reach the prescribed obstacle ratio $\rho$. This keeps difficulty roughly constant even when two copies of every station are spawned.

**Failure handling.** If any placement stage exhausts the pool of empty cells, or the validator rejects the grid, the attempt is aborted and restarted with a fresh $h, w$ sample. We cap retries at `max_attempts` (default 2000); empirically fewer than five attempts suffice for $\rho \le 0.3$.

**Complexity.** All placement operations are $O(hw)$ in the worst case (linear scans to collect empty cells), while validation runs a constant number of BFS passes, each $O(hw)$. Hence one successful attempt is $O(hw)$.

### A.3  LAYOUT VALIDATOR

We guarantee that every procedurally generated kitchen is *playable* by running a deterministic validator before training begins. The validator implements ten checks, ranging from basic grid sanity to cooperative reachability. A grid is accepted only if **all** checks pass.

**Listing 2** Overcooked Layout Generator

```python
def generate_random_layout(seed, params):
    rng = random.Random(seed)
    for attempt in range(params.max_attempts):
        h = rng.randint(*params.h_range)
        w = rng.randint(*params.w_range)
        grid = init_floor_with_border(h, w)

        # 1. Interactive tiles
        for sym in [GOAL, POT, ONION_PILE, PLATE_PILE]:
            if not place_random(grid, sym, rng.randint(1, 2), rng):
                break  # restart

        # 2. Extra walls to hit density
        n_target = round(params.wall_density * (h-2)*(w-2))
        n_add = n_target - count_unpassable(grid)
        if not place_random(grid, WALL, n_add, rng):
            continue  # restart

        # 3. Agents
        if not place_random(grid, AGENT, params.n_agents, rng):
            continue

        # 4. Validate
        if evaluate_grid(to_string(grid)):
            return to_string(grid)
```

**Notation.** Let $G$ be an $h \times w$ character matrix with symbols $\{W, X, O, B, P, A, \}$ for walls, delivery, onion pile, plate pile, pot, agent, and floor. Interactive tiles are $\mathcal{I} = \{X, O, B, P\}$, and unpassable tiles $\mathcal{U} = \mathcal{I} \cup \{W\}$.

**Validation rules.**

**R1** *Rectangularity* – all rows have equal length.

**R2** *Required symbols* – each of W,X,O,B,P,A appears at least once.

**R3** *Border integrity* – every outer-row/column tile is in $\{W\} \cup \mathcal{I}$.

**R4** *Interactivity access* – every tile in $\mathcal{I} \cup \{A\}$ has at least one 4-neighbour that is A or floor.

**R5** *Reachable onions* – at least one onion pile is reachable by some agent.

**R6** *Usable pots* – at least one pot is reachable *and* lies in the same connected component as a reachable onion.

**R7** *Usable delivery* – at least one delivery tile is reachable *and* lies in a component with a usable pot.

**R8** *Agent usefulness* – each agent can either interact with an object directly or participate in a hand-off (adjacent wall shared with the other agent's region).

**R9** *Coverage* – the union of agents' reachable regions touches every object family in $\mathcal{I}$.

**R10** *Handoff counter* – if one agent cannot reach all families, a wall tile adjacent to *both* regions exists, enabling item transfer.

Rules R5–R10 rely on two depth-first searches (DFS) from the agent positions. The DFS explores floor and agent tiles only; whenever it touches an interactive tile, that family is marked as "found." Let $\text{Reach}_k \subseteq [h] \times [w]$ denote tiles reached from agent $k$ ($k \in \{1, 2\}$).

**Algorithmic outline.** Listing 3 shows a condensed version of the validator.

**Complexity.** All checks are $O(hw)$ and require only two DFS traversals, thus one validation runs in time linear to the grid area and is negligible compared with policy learning.

**Listing 3** Condensed Layout Validator.

```python
def validate(grid_str):
    g = [list(r) for r in grid_str.splitlines()]
    h, w = len(g), len(g[0])

    # R1-R3 omitted for brevity ...

    # Depth-first search from a start cell
    def dfs(i, j, seen):
        if (i, j) in seen or g[i][j] in UNPASSABLE_TILES - {AGENT}:
            return
        seen.add((i, j))
        for di, dj in ((1,0),(-1,0),(0,1),(0,-1)):
            dfs(i+di, j+dj, seen)

    # Agents and family reachability
    a1, a2 = [(i, j) for i,r in enumerate(g)
                     for j,c in enumerate(r) if c == AGENT]
    reach1, reach2 = set(), set()
    dfs(*a1, reach1); dfs(*a2, reach2)

    # Helper: reachable(\mathcal{S}, reach)
    def any_reach(symbols, reach):
        return any(g[i][j] in symbols for i,j in reach)

    # R5-R7
    if not any_reach({ONION_PILE}, reach1|reach2):    return False
    if not any_reach({POT}, reach1|reach2):           return False
    if not any_reach({GOAL}, reach1|reach2):          return False

    # R8-R10 (usefulness & hand-off)
    def useful(reach_me, reach_other):
        # direct or shared-wall hand-off
        for i,j in reach_me:
            if g[i][j] in INTERACTIVE_TILES: return True
            if g[i][j] == FLOOR and any(
                (abs(i-i2)+abs(j-j2) == 1 and g[i2][j2] == WALL)
                for i2,j2 in reach_other):
                return True
        return False

    if not useful(reach1, reach2): return False
    if not useful(reach2, reach1): return False
    return True
```

**Practical impact.** In practice, fewer than 1% of generator attempts fail validation when wall-density $\rho \le 0.15$ and kitchen size $\ge 8 \times 8$. We therefore cap retries at 2000 without noticeable overhead.

## B  EXPERIMENTAL SETUP

### B.1  NETWORK ARCHITECTURE

All agents share the same actor–critic backbone, implemented in `Flax`. Two encoder variants are provided:

- **MLP** (default): observation tensor is flattened to a vector and passed through 2 fully-connected layers of width 128.
- **CNN**: three 32-channel convolutions with kernel sizes $5\times5$, $3\times3$, $3\times3$ feed a 64-unit projection, followed by a single 128-unit dense layer.

Common design knobs (controlled from the CLI) are:

- **Activation** (`relu` *vs.* `tanh`).
- **LayerNorm**: applied after every hidden layer when `use_layer_norm` is enabled.
- **Shared vs. Separate encoder**: with `shared_backbone` the two heads operate on a common representation; otherwise actor and critic keep independent trunks.
- **Multi-head outputs**: if `use_multihead` is set, each head holds a distinct slice of logits/values for every task ($\text{num\_tasks} = |\mathcal{T}|$). The correct slice is selected with a cheap tensor reshape.
- **Task-one-hot conditioning**: setting `use_task_id` concatenates a one-hot vector of length $|\mathcal{T}|$ before the actor/critic heads, mimicking "oracle" task identifiers used in many CL papers.

All linear/conv layers use orthogonal weight initialisation with gain $\sqrt{2}$ (or 0.01 for policy logits) and zero biases. The policy outputs a `distrax.Categorical`; the critic outputs a scalar.

## B.2 HYPERPARAMETERS

Table 5 lists settings that are *constant* across every experiment unless stated otherwise. Values match the `Config` dataclass in the training script.

Table 5: Fixed hyper-parameters. All experiments use dense reward shaping, two agents, and IPPO unless noted. CL coefficients $\lambda$ refer to the regularization strength passed to each method.

| Parameter | Value |
|---|---|
| *Optimization (PPO)* | |
| Activation | ReLU |
| Optimizer | Adam (Optax) |
| Adam $(\beta_1, \beta_2)$ | (0.9, 0.999) |
| Adam $\epsilon$ | $10^{-5}$ |
| Weight decay | none |
| Learning rate $\eta$ | $10^{-3}$ |
| LR annealing | linear $(10^{-3} \rightarrow 10^{-4})$ |
| Env. steps per task $\Delta$ | $10^8$ |
| Parallel envs | 2048 |
| Rollout length $T$ | 400 |
| Effective batch size | $2048 \times 400 = 819\,200$ |
| Updates per task | $\lfloor 10^8/819\,200 \rfloor = 122$ |
| Update epochs | 8 |
| Minibatches per update | 16 |
| Gradient steps per task | $122 \times 8 \times 16 = 15{,}616$ |
| Discount $\gamma$ | 0.99 |
| GAE $\lambda$ | 0.95 |
| PPO clip $\epsilon$ | 0.2 |
| Entropy coef. $\alpha_{\text{ent}}$ | 0.01 |
| Value-loss coef. $\alpha_{\text{vf}}$ | 0.5 |
| Max grad-norm | 1.0 |
| *Continual-learning specifics* | |
| Sequence length $|\mathcal{T}|$ | 20 (base sequence), repeated $r$ times |
| Reg. coefficient $\lambda$ | $10^{11}$ (EWC), $10^9$ (MAS), $10^7$ (L2) |
| Online EWC/MAS decay | 0.9 |
| Importance episodes / steps | 5 / 500 |
| Regularize critic / heads | No / No |
| AGEM Memory size | 100\,000 transitions |
| AGEM Sample size (per proj.) | 1024 |
| *Miscellaneous* | |
| Reward shaping horizon | $2.5 \times 10^6$ steps (linear to 0) |
| Evaluation interval | every 5 policy updates |
| Evaluation episodes | 10 |
| Random seeds | $\{1 .. 5\}$ |

Table 6: Field-of-view specification for the partially observable MEAL variant. Window size and directional extents scale with difficulty.

| Difficulty | Grid Size | Forward View | Side View | Rear View | Obs Window (H×W) |
|------------|-----------|--------------|-----------|-----------|------------------|
| Easy | 6–7 | 1 | 1 | 0 | 2×3 |
| Medium | 8–9 | 2 | 1 | 0 | 3×3 |
| Hard | 10–11 | 3 | 2 | 1 | 3×5 |

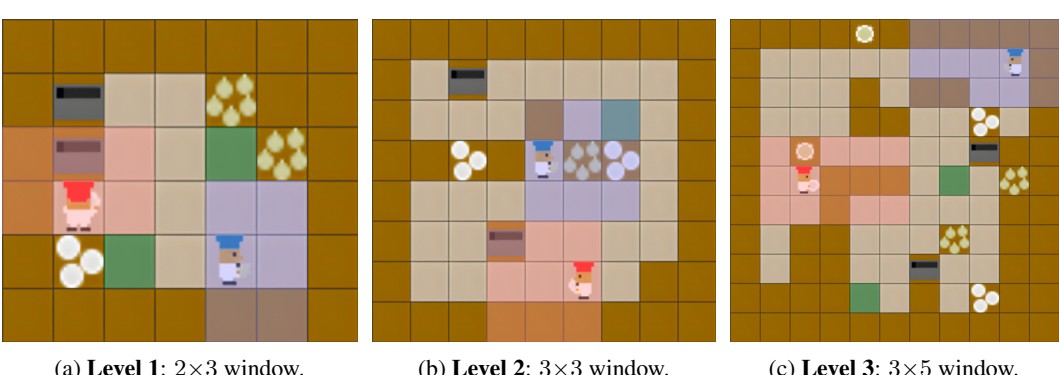

(a) **Level 1**: 2×3 window.  (b) **Level 2**: 3×3 window.  (c) **Level 3**: 3×5 window.

Figure 9: Egocentric observation windows by difficulty. Visibility grows with difficulty but remains partial, preserving the need for exploration and memory.

## C  PARTIALLY OBSERVABLE MEALS

To more closely mimic the constraints faced by real-world agents, we introduce a *direction-aware* egocentric observation setting. Each agent perceives a rectangular window centered on itself, with tiles outside this window masked. The window is anisotropic with respect to the agent's heading: we separate forward, side, and rear extents, which increase with difficulty (Table 6). This scaling is intentionally balanced with the overall environment design: as the grid size grows with difficulty, the perceptual window also expands to maintain a comparable challenge-to-information ratio. Consequently, the tasks become POMDPs, where exploration, memory (e.g., recurrent state), and implicit/explicit coordination provide tangible benefits. In particular, Level 1 removes rear context entirely, Level 2 extends the look-ahead by one tile, and Level 3 adds both longer look-ahead and rear visibility, reducing blind spots while preserving partial observability (Figure 9).

## D  DIFFICULTY LEVELS

Higher levels of difficulty in MEAL pose greater challenges for both learning and retention. As the grid size and obstacle density increase, the environment becomes more complex: interactable items are farther apart, and navigation paths are longer and more convoluted. This increases the number of steps required to complete a recipe. Not only does this make learning each task harder, but it also forces the agent to retain and execute longer action sequences to successfully complete a recipe. Higher-level layouts also add demands for plasticity and transfer. The larger layout space introduces greater variability between tasks, making it harder to reuse learned behavior. These factors collectively lead to lower performance as difficulty increases, as shown in Figure 10. Online EWC has a steady upward trend on Level 1, while on higher levels, the performance gap becomes more evident as the number of tasks increases.

## E  CURRICULUM LEARNING

In all training settings, agents consistently struggle on Level 3 tasks with large grids. Curriculum learning has been shown to improve final performance on difficult tasks by gradually increasing task complexity (Bengio et al., 2009; Narvekar et al., 2020; Portelas et al., 2020). We investigate whether

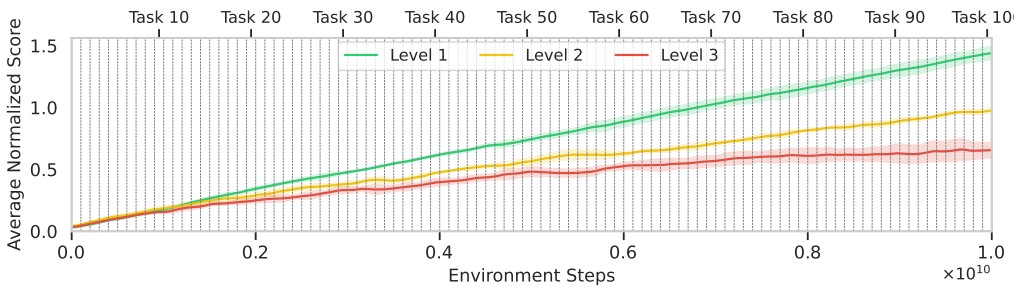

Figure 10: **Average Normalized Score** over the course of training Online EWC on a sequence of 100 generated tasks per difficulty level. Shaded regions indicate 95% confidence intervals across 5 seeds. The clear gap between levels shows the effectiveness of MEAL's naïve difficulty level design.

a simple difficulty-based curriculum can help agents better learn harder MEAL tasks under the same data budget. To this end, we design a curriculum sequence where each difficulty level contributes an equal number of tasks. Specifically, we sample 5 layouts each from Level 1 (easy), Level 2 (medium), and Level 3 (hard), and present them in ascending order of difficulty (layouts 1–5, then 6–10, then 11–15). As a baseline, we compare with a default sequence that trains on 15 hard (Level 3) layouts without any prior exposure to easier tasks. Performance is evaluated based on the normalized average score over the 5 tasks in the sequence of the respective difficulty.

The results in Table 7 show no statistically significant difference between the two strategies on Level 2, given the high variance. However, on Level 3, the curriculum strategy nearly doubles performance. A plausible explanation is that, under curriculum training, the agent first experiences 5 easy and 5 medium tasks, where it receives denser reward signals and more frequent successes. This exposure likely builds useful priors and stabilizes learning, improving adaptation to harder tasks later. In contrast, the default strategy trains only on hard tasks throughout the sequence, where exploration is more challenging and initial rewards are more difficult to obtain, leading to weaker performance overall.

Table 7: Curriculum vs. default training under an equal data budget. We report the average score over the task windows of the respective difficulty.

| Strategy | Medium (6–10) | Hard (11–15) |
|---|---|---|
| Default | $0.693 \pm 0.147$ | $0.328 \pm 0.238$ |
| Curriculum | $0.668 \pm 0.152$ | $\mathbf{0.653 \pm 0.181}$ |

## F  NETWORK PLASTICITY

Loss of neural network plasticity is a well-studied phenomenon in continual RL, where agents gradually become less able to adapt to new tasks as training progresses. A number of metrics have been proposed to characterize this effect, typically measuring how updates propagate through the network or how parameter sensitivity changes over time.

### F.1  METRICS

We follow Abbas et al. (2023); Dohare et al. (2024) and quantify **plasticity**, the ability to fit fresh data after many tasks, by three complementary metrics computed from the training reward.

**Notation.** For a single task let $r_t$ be the online reward at step $t \leq T$. A repetition experiment presents the same task $R$ times, so the trace splits into $R$ contiguous segments of equal length $L = T/R$. We smooth $r_t$ with a Gaussian kernel (bandwidth $\sigma$) and define the cumulative average

$$\bar{r}(t) = \frac{1}{t} \sum_{i=1}^{t} r_i, \qquad t = 1, \dots, L.$$

All metrics compare a later repetition $j > 0$ with the *baseline* repetition $j = 0$.

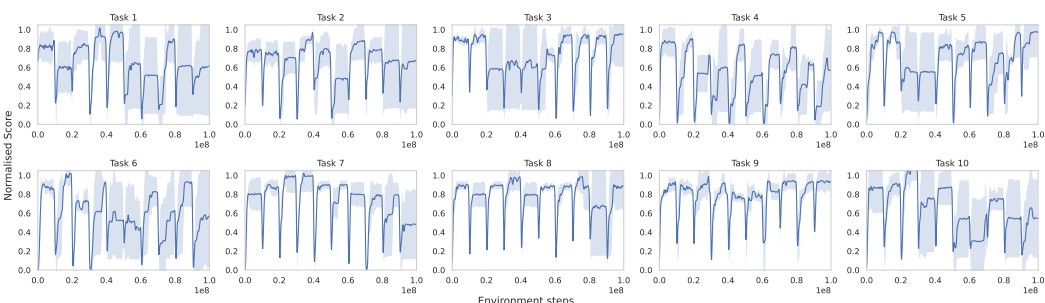

Figure 11: Training curves of FT across a Level 1 10-task sequence repeated ten times over 5 seeds.

Table 8: All sequence-averaged metrics for FT with 95% confidence intervals.

| Repeats | AUC-loss ↓ | Dormant Ratio ↓ | FPR ↑ | RAUC ↑ |
|---|---|---|---|---|
| 1 | $0.000 \pm 0.000$ | $0.408 \pm 0.003$ | $1.000 \pm 0.000$ | $1.000 \pm 0.000$ |
| 3 | $0.166 \pm 0.052$ | $0.428 \pm 0.006$ | $0.926 \pm 0.111$ | $0.901 \pm 0.114$ |
| 10 | $0.201 \pm 0.018$ | $0.509 \pm 0.022$ | $0.891 \pm 0.066$ | $0.872 \pm 0.070$ |

**AUC–loss.** Let $\mathrm{AUC}_j = \int_0^L \bar{r}_j(t)\, dt$. The capacity drop for repetition $j$ is

$$\mathrm{loss}_j \;=\; 1 - \frac{\mathrm{AUC}_j}{\mathrm{AUC}_0}, \qquad j = 1, \ldots, R-1, \tag{4}$$

where 0 indicates perfect retention. We report the mean of Eq. (4) over repetitions and seeds.

**Dormant Neuron Ratio.** Following Sokar et al. (2023), we also measure *dormancy*, the fraction of units that remain effectively inactive during training. Given hidden activations $h \in \mathbb{R}^{B \times H}$ for batch size $B$ and layer width $H$, we compute the mean absolute activation per unit $m = \frac{1}{B} \sum_{b=1}^{B} |h_{b,:}|$. Normalizing by the global mean $\bar{m} = \frac{1}{H} \sum_{j=1}^{H} m_j$, we obtain scores $s_j = m_j / (\bar{m} + \epsilon)$. A unit is considered *dormant* if $s_j \leq \tau$ for some threshold $\tau$ (we use $\tau = 0.01$). The Dormant Neuron Ratio is the fraction of dormant units, averaged across layers and seeds. Higher values indicate more inactive capacity, and hence reduced plasticity.

**Final-Performance Ratio (FPR).** With $p_j = \bar{r}_j(L-1)$ the plateau reward of repetition $j$,

$$\mathrm{FPR}_j \;=\; \frac{p_j}{p_0}, \qquad j = 1, \ldots, R-1, \tag{5}$$

so $\mathrm{FPR}_j > 1$ implies no loss, $\mathrm{FPR}_j < 1$ indicates degraded plateau performance.

**Raw-AUC Ratio (RAUC).** Using the *unsmoothed* running reward,

$$\mathrm{RAUC}_j \;=\; \frac{\mathrm{AUC}_j^{\mathrm{raw}}}{\mathrm{AUC}_0^{\mathrm{raw}}}, \qquad j = 1, \ldots, R-1, \tag{6}$$

which captures the total reward accumulated during learning. Higher values in Eq. (5)–Eq. (6) are better.

**Sequence-level aggregation.** For a task sequence of length $|\mathcal{T}|$ we compute the per-task means of (4)–(6) and average across tasks, yielding a single global score per repetition count $R$.

### F.2 TRAINING CURVES

Figure 11 plots the mean normalized score of the fine-tuning (FT) baseline over ten repetitions. Performance on Tasks 8 and 9 remains virtually unchanged, indicating little to no plasticity loss. In contrast, Tasks 1, 2, 6, and 10 show a clear degradation: the agent fails to recover the score achieved during the first repetition, illustrating a pronounced loss of plasticity.

Table 9: Homogeneous vs. heterogeneous (designated roles) 2-agent training results over Level 1 20-task generated sequences using shared rewards and IPPO in combination with EWC.

| Setting | $\mathcal{A}\uparrow$ | $\mathcal{F}\downarrow$ | $\mathcal{FT}\uparrow$ |
|---|---|---|---|
| Homogeneous | $\mathbf{0.90 \pm 0.04}$ | $\mathbf{0.01 \pm 0.01}$ | $\mathbf{0.20 \pm 0.08}$ |
| Heterogeneous | $0.68 \pm 0.09$ | $0.03 \pm 0.02$ | $-0.05 \pm 0.09$ |

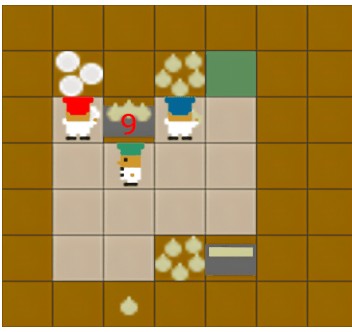 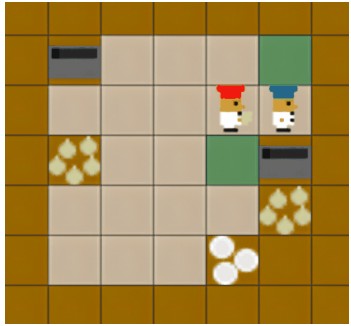 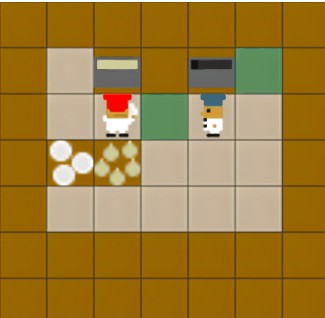

(a) **Single-pot fixation**. All agents are clustered around a single pot, waiting for it to finish cooking, while ignoring a ready soup in the bottom pot.

(b) **Deadlock**. The red agent tries to place an onion into the pot, but is blocked by the blue agent, who cannot move aside.

(c) **Role collapse.** One agent completes the pipeline solo while the other wanders or idles. The policy settles on a local minimum.

Figure 12: Qualitative failure modes observed in Overcooked. All behaviors stem from inadequate coordination, limited exploration, or insufficient role allocation.

# G  DESIGNATED ROLES

In Overcooked, agents are identical in their capabilities and attributes. However, in many real-world scenarios, autonomous agents either 1) possess different physical properties or 2) are functionally identical but are expected to fulfill distinct, complementary roles to cooperate effectively for a common goal. To capture this dimension in MEAL, we design a heterogeneous agent setting with **designated roles**.

In this variant, two agents are randomly assigned one of the two predefined roles at the start of each task: **chef** and **waiter**. The chef is responsible for preparing the soup by loading onions into the pot, but cannot pick up plates. The waiter handles dish delivery but cannot pick up onions. This enforces complementary capabilities, meaning neither agent can complete the full recipe alone, meaning that successful catering requires coordinated role execution and adaptation. Note that the roles are sampled per task and may switch across tasks, making continual learning essential.

We evaluate this setting over 20-task Level 1 sequences using EWC with IPPO under shared rewards. Table 9 compares the heterogeneous setup to the default homogeneous setting. We observe a clear performance drop in the role-restricted setting, as throughput decreases when agents are limited to certain actions and cannot flexibly switch between tasks. Another factor is asymmetric step costs: in many layouts, loading the pot with 3 onions takes more steps than a single plate-and-deliver trip, making the chef the throughput bottleneck. Generalization also suffers as agents struggle to transfer knowledge when their roles change across tasks, since skills learned in one role do not apply to the other. This role-switching dynamic further exacerbates forward transfer challenges in continual learning.

# H  COMMON PITFALLS

Despite shared rewards and simple layouts, learned policies frequently fall into recurring failure modes that throttle throughput and coordination. Figure 12 illustrates three such patterns we observe consistently across layouts and levels.

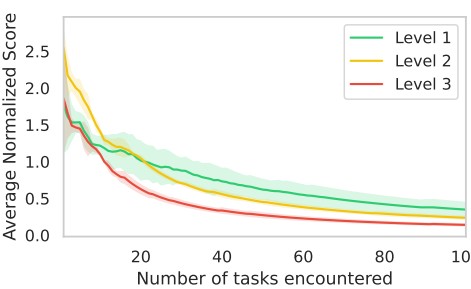 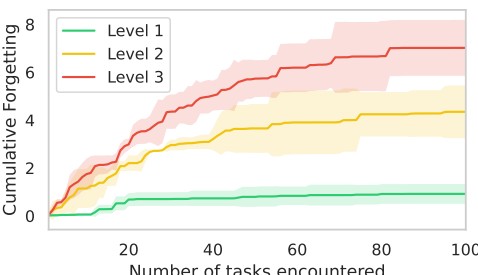

Figure 13: EWC over 100 tasks. **Left**: gradual decline in average score as more tasks are encountered. **Right**: on higher levels, forgetting increases more rapidly.

## I    100-TASK SEQUENCES

For the extensive analysis in this paper (e.g., partial observability, model component ablations, different numbers of agents), we use 20-task sequences to maintain a fast evaluation loop. However, a core aspect of MEAL is its ability to support continual learning at scale. To demonstrate this capability, we evaluate *100-task* sequences. To the best of our knowledge, such an extensive continual RL task sequence has not previously been reported in the literature.

For these runs, we reduce the evaluation frequency to once every 25 policy updates, yielding fewer data points. We evaluate EWC using the same training configuration as in our main experiments. Figure 13 shows that performance gradually declines as the number of tasks grows. The performance gap across difficulty levels is especially pronounced for forgetting. Executing 10 billion environment steps (plus policy updates) for 100 tasks required $\sim 4$ hours on a single GPU.

### I.1    A CASE STUDY: EWC VS. ONLINE EWC

EWC accumulates importance over *all* past tasks and penalizes drift along high-Fisher directions with a fixed quadratic. Online EWC maintains a *running*, exponentially decayed Fisher, emphasizing recent tasks and relaxing old constraints. Both use the same heads, meaning that the penalty acts on the shared trunk. When layouts are small, not only are the tasks easier to learn, but the same features are more likely to work across tasks. Strong anchoring preserves those features, curbing forgetting and yielding a higher average score. The stability–plasticity trade-off is favorable because plasticity demands are modest. This trade-off is visualized in Figure 14, where Online EWC demonstrates higher plasticity at the cost of increased forgetting, while EWC excels in stability but struggles to adapt on Level 3. The cumulative Fisher penalty pays off on small Level 1–2 layouts, but underfits on Level 3 since harder layouts demand larger representation shifts. By contrast, Online EWC uses a decayed Fisher that down-weights older tasks and manages to keep enough plasticity to learn the new layouts. Level 3 forces longer paths, bottlenecks, and role specialization, which require larger representational updates. EWC's cumulative constraints over-tighten the trunk and slow adaptation, while Online EWC's decay frees capacity for those shifts, so it learns the hard tasks more effectively. The multiple output heads alone are not enough. They isolate outputs, but the penalty sits on the shared backbone. When the trunk needs to be rewired for new Level 3 tasks, EWC resists too much, while Online EWC allows it more. Moreover, credit assignment is noisier on Level 3 due to sparser effective signals and longer horizons. A single, stale Fisher snapshot can misdirect EWC's penalty. The rolling estimate in Online EWC smooths that noise and tracks the current regime more closely.

Scaling up the number of tasks from 20 to 100 amplifies the behavioral difference between these methods. As shown in Figure 15, standard EWC quickly saturates: its performance plateaus around 20 tasks, while Online EWC continues to improve throughout the sequence. The difference arises from how the two methods accumulate and apply their regularization terms. EWC optimizes the loss

$$\mathcal{L}_{\text{EWC}} = \mathcal{L}_{\text{task}} + \frac{\lambda}{2} \sum_{t=1}^{k-1} \sum_{i} F_{t,i} \left( \theta_i - \theta_{t,i}^* \right)^2, \tag{7}$$

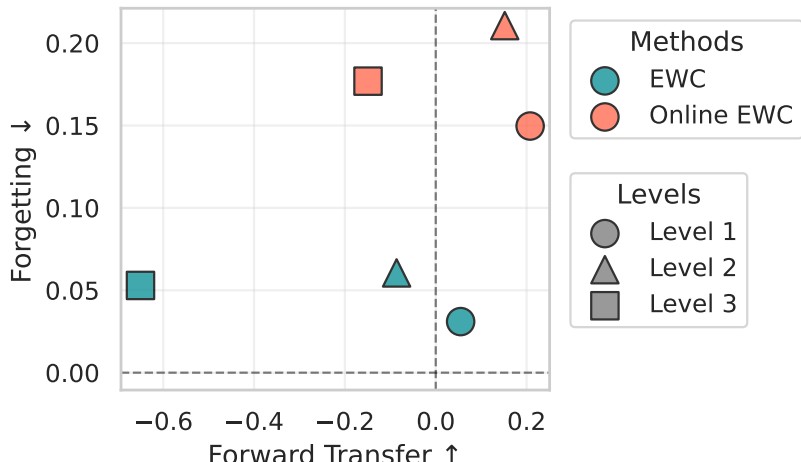

Figure 14: Comparison of EWC and Online EWC across all difficulty levels on 20-task sequences, evaluated in terms of forward transfer and forgetting. Each point denotes a method's performance at a given level. Online EWC consistently exhibits higher plasticity (less-negative or positive, particularly at Level 3, while EWC achieves notably lower forgetting on all levels.

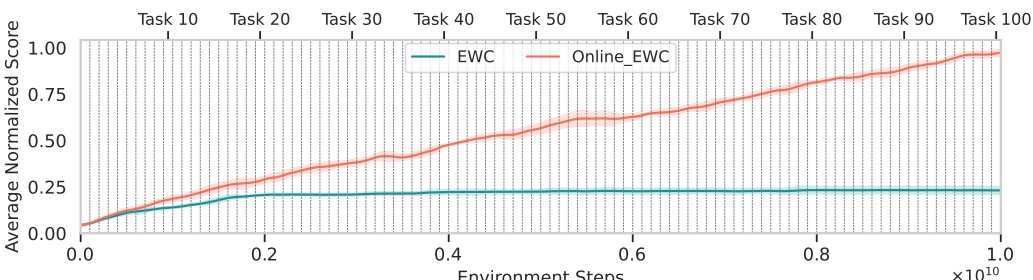

Figure 15: Average normalized performance of EWC and Online EWC over a 100-task sequence (Level 2). The standard EWC variant continuously accumulates regularization terms from all previous tasks, leading to excessive constraint and early performance saturation around 20 tasks. In contrast, Online EWC compresses past information with a decay factor, maintaining plasticity and achieving sustained learning throughout the sequence.

where each Fisher matrix $F_t$ captures parameter importance after task $t$. The regularizer grows with every task, anchoring the network more tightly to older solutions. Consequently, plasticity decays over time, and adaptation to new tasks becomes progressively harder. Online EWC instead compresses past information through an exponentially decayed Fisher:

$$F_{\text{online}}^{(k)} = \gamma F_{\text{online}}^{(k-1)} + F_k, \tag{8}$$

where the decay factor $\gamma \in (0,1)$ controls how quickly the influence of older tasks fades to restore the capacity for new ones. This running Fisher approximation is then used to define a single consolidated quadratic penalty:

$$\mathcal{L}_{\text{Online EWC}} = \mathcal{L}_{\text{task}} + \frac{\lambda}{2} \sum_i F_{\text{online},i}^{(k)} \left( \theta_i - \theta_{k,i}^* \right)^2. \tag{9}$$

Over long sequences, these mechanisms diverge sharply: standard EWC eventually over-regularizes, effectively freezing the shared backbone. The model retains early knowledge but cannot repurpose

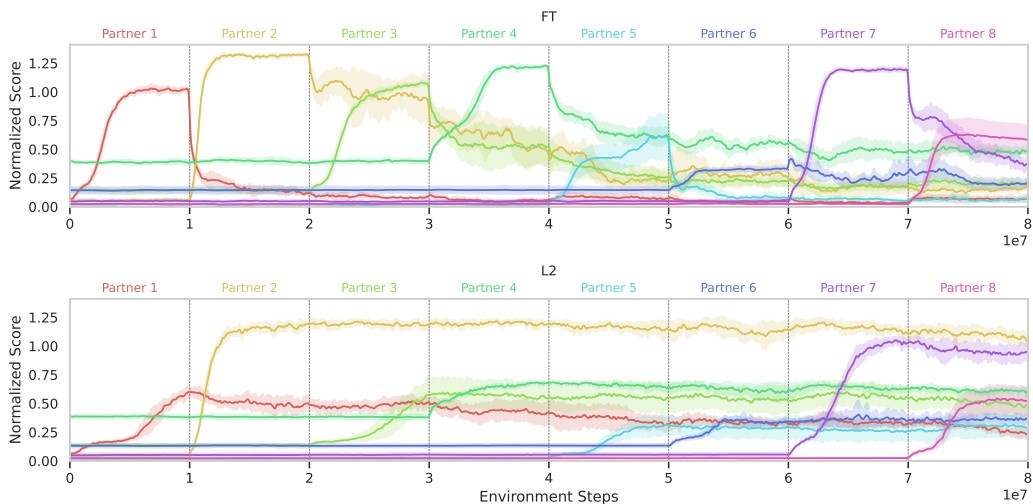

Figure 16: Evaluation curves of adapting to 8 diverse partners.

features as the environment shifts. Online EWC's decayed Fisher avoids this over-constraining and sustains meaningful adaptation even after dozens of tasks. The continued improvement across 100 tasks illustrates that certain dynamics in continual reinforcement learning only emerge over *long horizons*, where saturation and drift become clear.

This case study reinforces one of the core motivations behind **MEAL**: short sequences often fail to reveal such discrepancies. We therefore urge the continual RL community to focus on longer streams of tasks, where stability–plasticity trade-offs are more likely to truly emerge.

## J   CONTINUAL PARTNER ADAPTION

MEAL enables the generation of diverse partner policies, allowing continual learning methods to be evaluated not only across layouts but also across sequences of partners, e.g., $\mathcal{T} = (\pi_p^0, \ldots, \pi_p^L)$, where $L$ is the sequence length. To this end, we aim to generate partner policy sequences that are maximally diverse in their behaviour.

As described in Section 4.4, we use (i) hardcoded strategies (random, static), (ii) planning-based agents (onion-only, plate-only, and a human-like planner with stochastic task selection), and (iii) populations trained with best-response diversity (Rahman et al., 2023, BRDiv), which maximizes self-play performance while minimizing cross-play compatibility.

BRDiv populations in particular yield highly incompatible strategies. Coordinating with a new BRDiv partner typically requires learning behaviours that differ substantially from those seen before, making them a strong testbed for continual adaptation. In our experiments, we train BRDiv populations with a size of three, a cross-play weight of $1.0$, in $64$ parallel environments, using simple MLP policies.

Planning-based agents follow fixed strategies that learned policies rarely adopt. The onion-only agent collects onions and, with probability $p_{\text{onion-counter}} = 0.1$, places them on counters instead of pots. The plate-only agent collects plates and delivers dishes. With probability $p_{\text{plate-counter}} = 0.1$ it places the plate on a counter instead of plating a soup. The human-like planner follows simple heuristics: it prioritizes filling pots, but if no pot is free, it collects a plate and delivers a soup. With probability $0.1$, it may place either onions or plates on counters instead.

For the partner-adaptation experiments, we fix the schedule as follows: the ego agent first encounters the three BRDiv partners, followed by the human-like planner, then the onion-only, plate-only, random, and static agents. Unlike the layout-adaptation experiments, we keep the environment fixed to the `cramped_room` layout from the original Overcooked repository (Carroll et al., 2019), which is particularly sensitive to variations in partner behaviour.

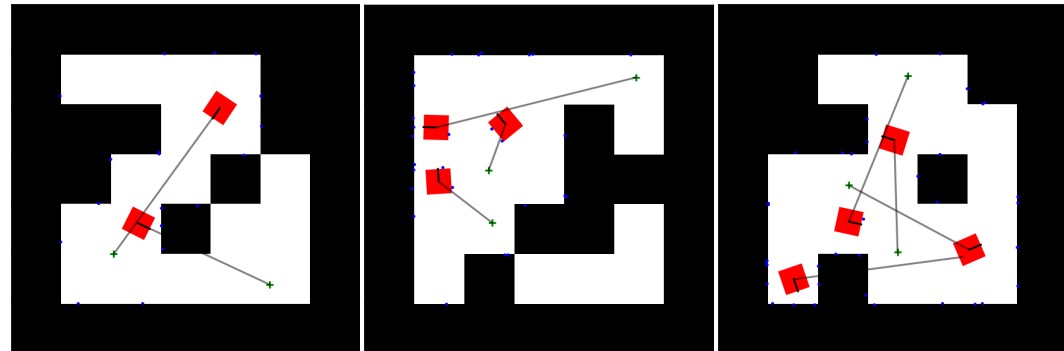

Figure 17: Example **JAxNav** environments on a 7×7 grid with 2–4 agents. Dark cells denote walls/obstacles, and light cells denote free space. Agents are depicted as red squares with orientation markers. Goal locations are marked with crosses. Thin lines show the direct vector from each agent to its goal. Blue dots visualize lidar sensor returns.

In our partner-adaptation experiments, we use a fixed schedule: the ego agent is first exposed to the three BRDiv policies, followed by the human-like planner, then the onion-only, plate-only, random, and static agents. Instead of using the *Meal Generator* (Section 4.2) to generate an array of different layouts, we adopt the `cramped_room` layout from the original Overcooked repository Carroll et al. (2019). In this layout, the ego agent has been shown to be susceptible to variations in partner behaviour Ruhdorfer et al. (2025b).

Table 10 compares the performance between naive fine-tuning(FT) and L2-regularization. FT manages to forget less in this setting and obtains a higher average score than when adapting to different layouts (Table 2. The continual evaluation curves in Figure 16 show that when the ego agent is exposed to a new fixed partner policy, it can still coordinate to some extent with previous partners, although rapidly adapting its policy to align with the new partner. In contrast, such transfer is largely absent when the challenge comes from adapting to a new layout. Contrary to FT, L2 performs worse in this setting compared to layout adaptation, forgetting more and delivering fewer soups on average. Improved retention comes at the cost of adapting less freely and achieving less total soup deliveries during training.

Table 10: Continual learning metrics for partner adaptation across 8 diverse partners in the `cramped_room` layout.

| Method | $\mathcal{A}\uparrow$ | $\mathcal{F}\downarrow$ |
|---|---|---|
| FT | $0.272_{\pm 0.03}$ | $0.707_{\pm 0.02}$ |
| L2 | $\mathbf{0.563}_{\pm 0.04}$ | $\mathbf{0.172}_{\pm 0.02}$ |

## K  JAxNav

While in the scope of this paper, MEAL is centered around Overcooked, we wish to demonstrate that the continual learning framework is not restricted to a single domain. To this end, we incorporate JAxNav Rutherford et al. (2024a), a navigation-based continuous multi-agent environment. JAxNav introduces different challenges: agents must reach their assigned goal locations relying on local lidar observations while avoiding collisions with walls and each other. Some layouts contain narrow bottlenecks where two agents can not pass simultaneously, requiring one to explicitly give way to the other(s).

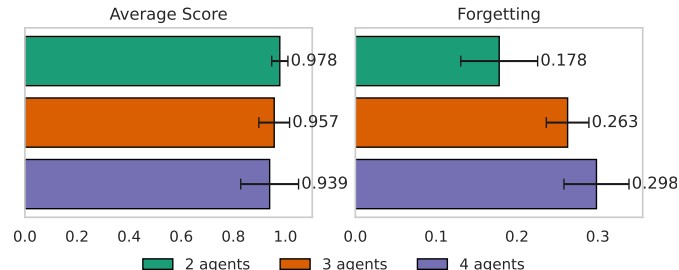

Figure 18: Online EWC performance on JAxNav over 20-task sequences. Forgetting increases steadily with additional agents, while average score degrades more mildly.

To create continual learning task sequences, we rely on JAxNav's built-in randomized layout generator. For our experiments, we use 7×7 grids with an obstacle fill ratio of 0.3. Unlike Overcooked, where we derive continual learning metrics from the normalized soup delivery score, JAxNav allows

us to compute these metrics directly from the environment's raw returns. We keep all training settings identical to OverCooked, including the MLP encoder, PPO algorithm, Adam optimizer, hyperparameters, and evaluation schedule. We run 20-task sequences over 5 seeds and evaluate Online EWC with 2–4 agents. Figure 17 depicts example environments used in our experiments.

We can observe a slight downward trend in performance in Figure 18 when increasing the number of agents. Importantly, the increase of forgetting is more notable, meaning that tasks with more agents are harder both to learn and to remember. This once again reflects a key point of our work: continual learning gets more difficult as the number of interacting agents increases. The addition of JAXNAV shows that MEAL naturally extends to other JAX-based environments while preserving its high-throughput training pipeline.

## L USE OF LLMS

We used large language models (LLMs) exclusively to aid in polishing the language and improving the clarity of presentation. No part of the research design, experiments, or analysis was generated or influenced by LLMs.

## M EXTENDED RESULTS

In this section, we provide additional experimental results. Tables 11, 12, 13 add 95% confidence intervals to the main baseline results. Figures 19 and 20 show performance curves of higher levels. Figure 21 depicts the per-task evaluation curves of Level 1. Figure 22 illustrates forward transfer.

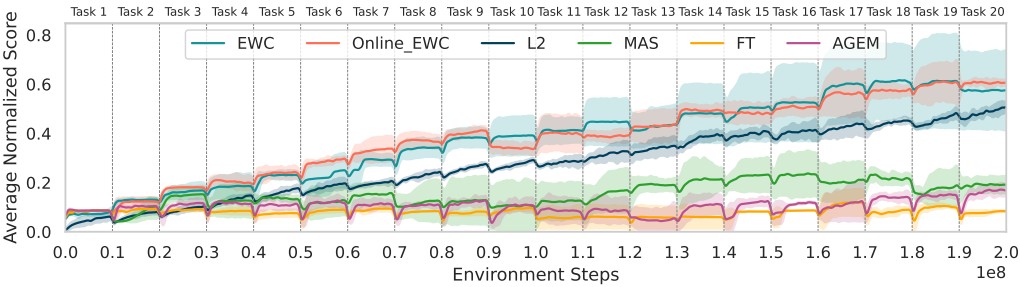

Figure 19: **Average Normalized Score** curves on Level 2.

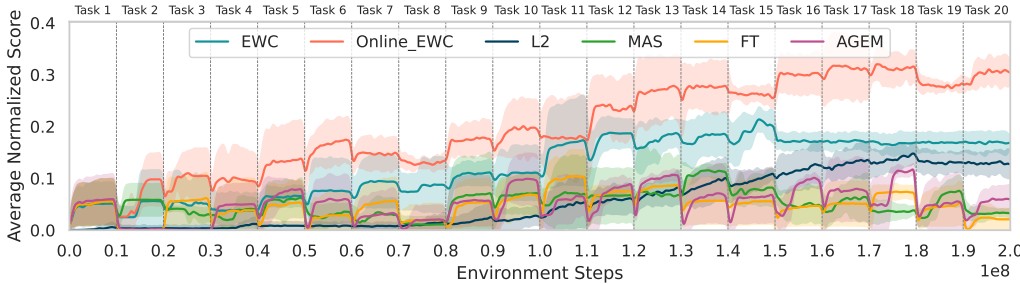

Figure 20: **Average Normalized Score** curves on Level 3.

Table 11: Level 1 baseline results with confidence intervals.

| Method | $\mathcal{A}\uparrow$ | $\mathcal{F}\downarrow$ | $\mathcal{FT}\uparrow$ |
|---|---|---|---|
| FT | $0.048_{\pm 0.00}$ | $0.946_{\pm 0.00}$ | $0.201_{\pm 0.03}$ |
| EWC | $\mathbf{0.839}_{\pm 0.03}$ | $\mathbf{0.012}_{\pm 0.01}$ | $0.055_{\pm 0.06}$ |
| Online EWC | $0.769_{\pm 0.09}$ | $0.062_{\pm 0.05}$ | $\mathbf{0.208}_{\pm 0.03}$ |
| MAS | $0.281_{\pm 0.07}$ | $0.302_{\pm 0.08}$ | $-0.233_{\pm 0.03}$ |
| L2 | $0.753_{\pm 0.02}$ | $0.018_{\pm 0.00}$ | $-0.199_{\pm 0.09}$ |
| AGEM | $0.204_{\pm 0.05}$ | $0.678_{\pm 0.04}$ | $0.125_{\pm 0.10}$ |

Table 12: Level 2 baseline results with confidence intervals.

| Method | $\mathcal{A}\uparrow$ | $\mathcal{F}\downarrow$ | $\mathcal{FT}\uparrow$ |
|---|---|---|---|
| FT | $0.041_{\pm 0.01}$ | $0.944_{\pm 0.00}$ | $0.065_{\pm 0.02}$ |
| EWC | $\mathbf{0.604}_{\pm 0.21}$ | $\mathbf{0.027}_{\pm 0.01}$ | $-0.086_{\pm 0.34}$ |
| Online EWC | $0.585_{\pm 0.03}$ | $0.100_{\pm 0.04}$ | $\mathbf{0.152}_{\pm 0.05}$ |
| MAS | $0.155_{\pm 0.09}$ | $0.356_{\pm 0.06}$ | $-0.355_{\pm 0.06}$ |
| L2 | $0.496_{\pm 0.02}$ | $0.059_{\pm 0.00}$ | $-0.527_{\pm 0.04}$ |
| AGEM | $0.117_{\pm 0.01}$ | $0.801_{\pm 0.02}$ | $-0.083_{\pm 0.07}$ |

Table 13: Level 3 baseline results with confidence intervals.

| Method | $\mathcal{A}\uparrow$ | $\mathcal{F}\downarrow$ | $\mathcal{FT}\uparrow$ |
|---|---|---|---|
| FT | $0.010_{\pm 0.02}$ | $0.947_{\pm 0.05}$ | $-0.157_{\pm 0.20}$ |
| EWC | $0.178_{\pm 0.02}$ | $\mathbf{0.091}_{\pm 0.08}$ | $-0.650_{\pm 0.13}$ |
| Online EWC | $\mathbf{0.306}_{\pm 0.00}$ | $0.144_{\pm 0.01}$ | $\mathbf{-0.149}_{\pm 0.15}$ |
| MAS | $0.034_{\pm 0.01}$ | $0.450_{\pm 0.09}$ | $-0.542_{\pm 0.20}$ |
| L2 | $0.127_{\pm 0.03}$ | $0.096_{\pm 0.00}$ | $-0.827_{\pm 0.04}$ |
| AGEM | $0.037_{\pm 0.02}$ | $0.861_{\pm 0.01}$ | $-0.169_{\pm 0.18}$ |

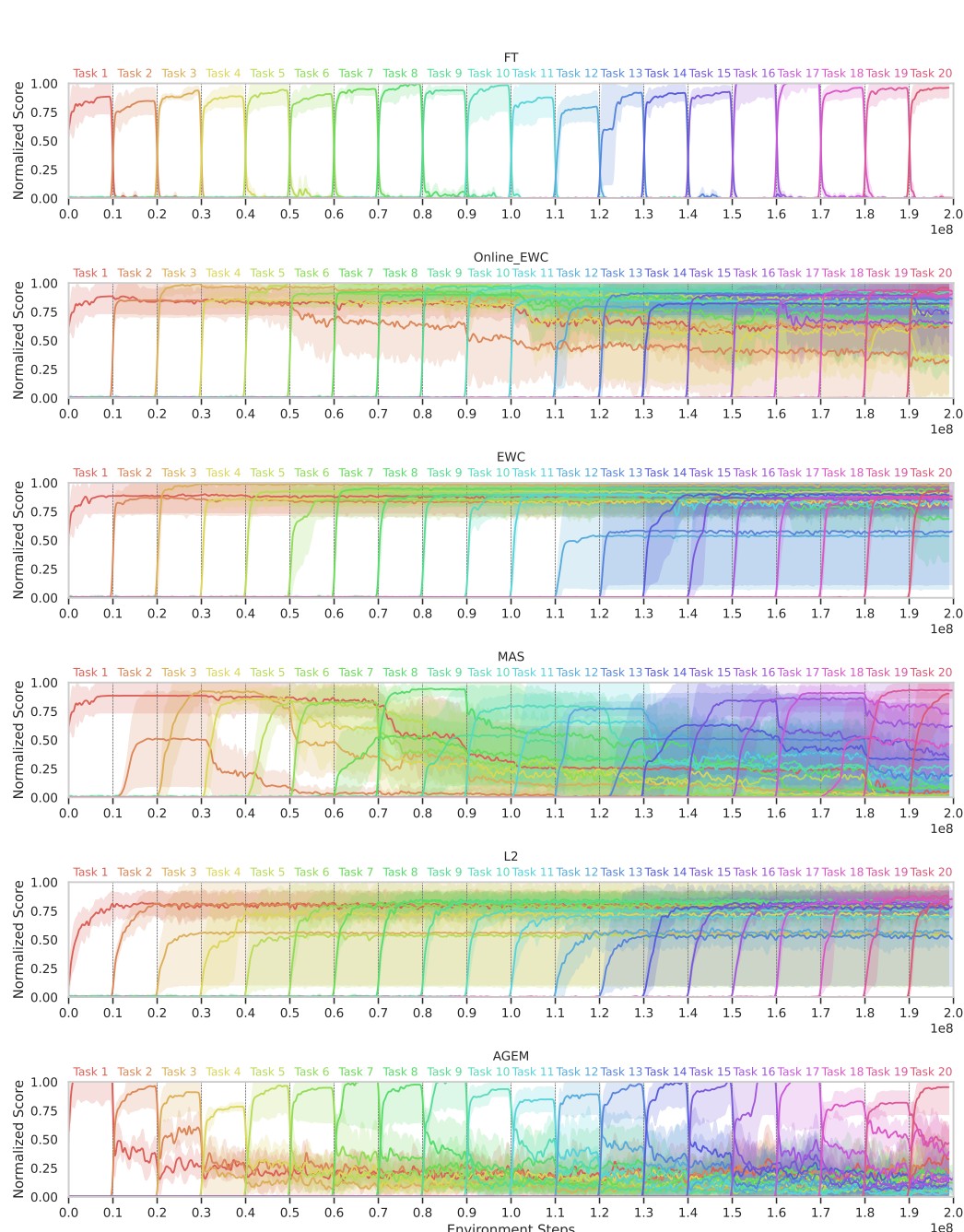

Figure 21: The **evaluation curves** of Level 1 illustrate the extent of forgetting across tasks. FT suffers from clear catastrophic forgetting: once the agent transitions to a new task, performance on the previous task collapses immediately. EWC and L2 display near-perfect retention.

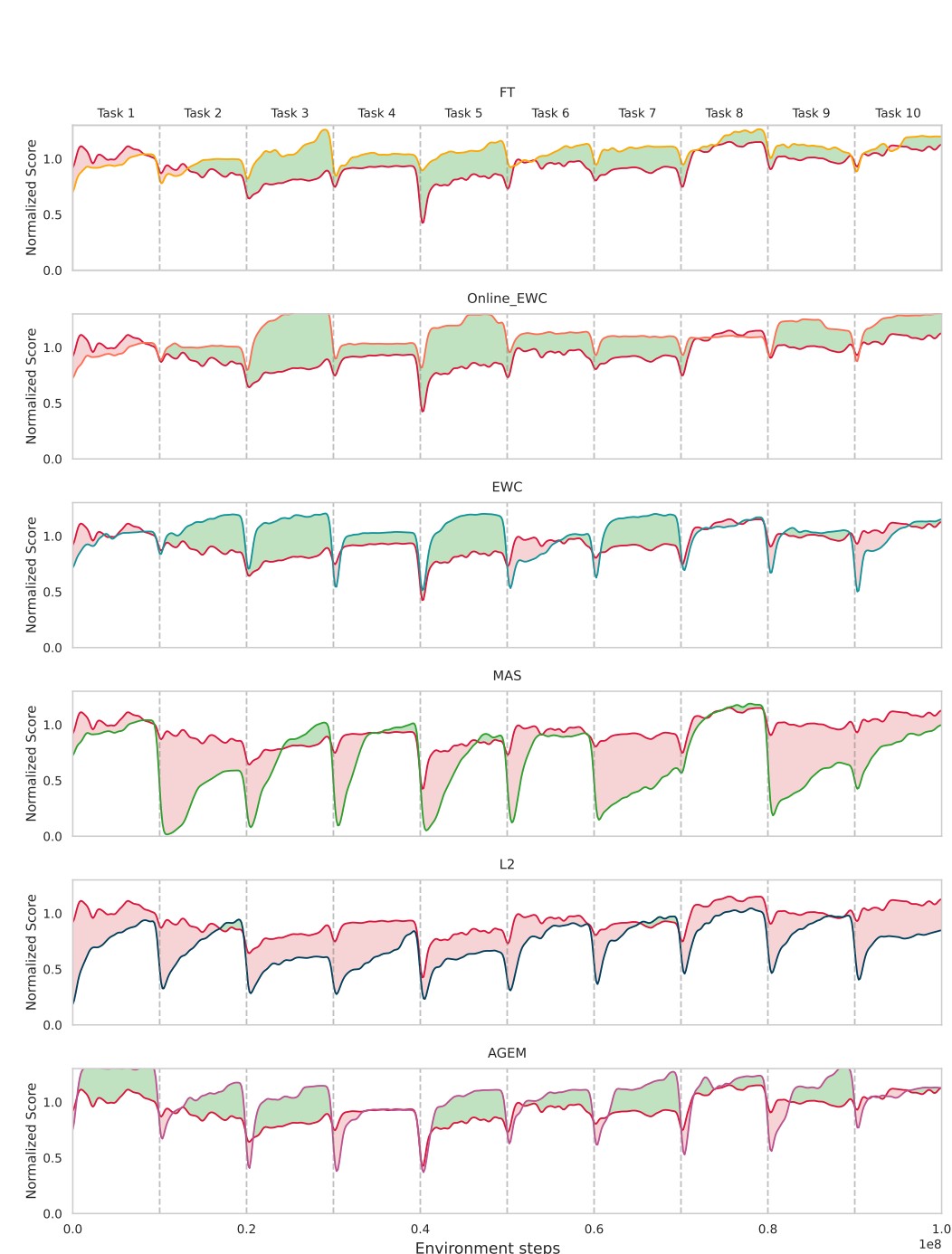

Figure 22: **Forward transfer** on Level 1. The green shaded areas depict positive transfer compared to the IPPO baseline, and the red shaded areas show negative transfer.

