# OpenReview forum: "MEAL: A Benchmark for Continual Multi-Agent Reinforcement Learning"
_ICLR.cc/2026/Conference — Submitted to ICLR 2026_

### Official Review · Reviewer_tMtN · 2025-10-28

**Soundness:** 2
**Presentation:** 3
**Contribution:** 2
**Rating:** 4
**Confidence:** 4

**Summary:**

The authors propose a benchmark for a Continual Multi-agent Reinforcement Learning. The benchmark environment is based on the “Overcooked” environment, written in JAX to enable massive learning time speedup, which in turn allows for the testing of more tasks. Authors also implemented several multi-agent RL methods and various methods used in the Continual RL setting, and evaluated their implementations across multiple configurations of the environment.

**Strengths:**

* The paper is clear, well-written, and coherent.
* Due to the nature of the Continual RL domain, there is a need for speeding up the environments, as training has to span multiple tasks, which drives training times to be long. The speed achieved by MEAL seems to be impressive, and allows for in-depth algorithm analysis with a data-rich setup
* Authors included several ablations and smaller experiments, which help to paint a more complete picture of how good current baselines are, and what challenges future algorithms will have to address.

**Weaknesses:**

I have several smaller issues with the paper that I believe should be addressed by the authors.

* The authors claim that MEAL is the first Continual Multi-agent RL benchmark that uses JAX for end-to-end training. I have found another GitHub project that appears to address the same setting, also utilizing an overcooked environment: [https://github.com/aialt/overcooked-jax](https://github.com/aialt/overcooked-jax). This project appears to differ significantly from the author-provided code in the anonymous GitHub repository, so I assume it is not the same project. Could the authors clarify what the differences are between this project and their contribution, and how the novelty claim holds in the presence of that other repository?
* My understanding is that JaxMARL([https://github.com/FLAIROx/JaxMARL](https://github.com/FLAIROx/JaxMARL)) already provides an overcooked environment for Multi-agent RL setup, and the authors acknowledge that their contribution is built on top of that project. Effectively, the benchmark provided is simply a level generator for the JaxMARL overcooked environment. In my opinion, while this benchmark can be used to assess different baselines and is useful, it should provide more than just a level generator for one environment. I would love to see at least one more environment.
* In section 4.5 authors write:
  “We therefore normalize the delivery count by the optimal cook-deliver cycle for a single agent on any given task”.
  I see the value in that metric, and indeed, in many scenarios, it can provide meaningful information on whether RL agents can effectively cooperate. The level validator that authors use rejects any level that cannot be solved by a single agent. This makes sense in the context of the above metric, because if a level is unsolvable by a single agent, the metric loses its meaning. But I believe that authors should include another level generator that forces the cooperation between the agents, i.e., the level is unsolvable by a single agent. This could be treated as a separate environment or incorporated as a level that is generated as part of the benchmark. Forced cooperation appears to be an important feature in a benchmark like this, and I don’t see any reason not to include it.
* The paper is well-written and clear for the most part, but I think that in section 4.1, the authors should add 1-2 sentences of explanation of the goal of the overcooked environment, and what is the meaning of onions, plates, deliveries, etc. Alternatively, authors could place this explanation in the appendix and refer to it; however, either way, the reader's understanding of the basic environment dynamics should not require them to look beyond the reviewed paper.
* The fact that the main experiments use multi-headed architectures (one head for each task) is mentioned for the first time in section 5.2 \- Ablation Study. I think that this should be mentioned earlier (section 5.1 perhaps?), as it is an important (although natural) choice.
* In the abstract, the authors claim:
  “MEAL leverages JAX for GPU acceleration, enabling continual learning across sequences of 100 tasks on a standard desktop PC in a few hours.”
  However, most tasks used in the paper are 20\. There should be at least one experiment in which authors push the benchmark to the limits and use 100 tasks, which they claim is possible.
* In Appendix B2, the hyperparameter for parallel envs is only 16\. This seems peculiar, as the primary speedup from using JAX is the parallelization across a number of environments. In similar works that use JAX for fast benchmarks, the authors typically use environments in the order of thousands or tens of thousands (for example, in the JaxMarl blog post, the authors compare 100, 1k, and 10k environments). What is the reason for such a small number? Have the authors tested larger numbers to achieve higher GPU utilization, and consequently greater learning speed?

**Questions:**

Although I don’t think it is necessary in this work, I would like to see a study of how scaling of the network sizes impacts performance. In the current version, the authors use 2x128 MLP networks, which are small even for RL algorithms. Especially in light of recent scaling results (for example, https://arxiv.org/abs/2405.16158, https://arxiv.org/abs/2410.09754), I think it is important to leverage the speed of JAX to provide some indication of whether scaling the networks achieves any performance gain.

---

> ### Author Response · Authors · 2025-11-20
>
> **W1 Same Setting Repository**
> > [...] https://github.com/aialt/overcooked-jax. [...]
>
> We are unable to access this repository. It appears to be private or deleted.
>
> **W2 “*Just a Level Generator*”**
>
> > My understanding is that JaxMARL(https://github.com/FLAIROx/JaxMARL) already provides an overcooked environment for Multi-agent RL setup, and the authors acknowledge that their contribution is built on top of that project. Effectively, the benchmark provided is simply a level generator for the JaxMARL overcooked environment.
>
> While referring to MEAL as “*simply a level generator*” understates the contribution, in principle, the reviewer is not wrong. Then again, under this framing, aren't all RL benchmarks just a bundle of levels (generated or hand-crafted)? MEAL's layout generator is undoubtedly a crucial factor to facilitate these long CL sequences, and includes several nontrivial design choices: scalable difficulty, the normalized soup metric, and layout validation. Beyond this, MEAL required quite some engineering efforts to support N-agents, partial observability, parallelized periodic evaluation during training, and implementations of CL baselines - all end-to-end in JAX.
>
> > In my opinion, while this benchmark can be used to assess different baselines and is useful, it should provide more than just a level generator for one environment. I would love to see at least one more environment.
>
> This would shift the scope of the paper quite a bit. Most aspects are currently devoted to Overcooked, from the acronym to the *Procedural Kitchen Generator*. We made a conscious choice to prioritize depth over breadth. Rather than scratching the surface of a wide array of environments, we provide extensive analysis about a single one, including the number of agents, difficulty levels, curriculum learning, role restrictions, network plasticity, partial observability, adaptation to held-out partners, reward strategies, and component ablations.
>
> That being said, we acknowledge the reviewer’s concern about the lack of breadth. To address this, we decided to include **JaxNav** from JaxMARL. We deem its purpose to primarily show that MEAL’s CL pipeline is not restricted to Overcooked, and can be extended to other (JaxMARL) environments. We intend to frame this additional environment as a proof-of-concept extension while keeping the core focus of the paper on Overcooked. We shall briefly mention it in the main text and dedicate a section in the appendix with environment specifications, task-generation details, metrics, and results.
>
> We keep our setup identical to the default JaxNav configuration. For each task, we rely on JaxNav’s built-in layout generator with the default grid size of 7x7 and obstacle fill ratio of 0.3, producing a randomized layout per task. We use the same evaluation protocol as MEAL, except that we compute CL metrics from "*Success*" instead of delivered soup. Success [0, 1] denotes the proportion of goals reached across episodes. Training settings remain unchanged (MLP encoder, PPO, optimizer, hyperparameters, etc). We evaluate Online EWC on 20-task sequences with 2–4 agents.
>
> | Agents     | 𝒜↑              | 𝓕↓              |
> |------------|------------------|------------------|
> | 2 Agents   | **0.978 ±0.03**  | **0.178 ±0.05**  |
> | 3 Agents   | 0.957 ±0.06      | 0.263 ±0.03      |
> | 4 Agents   | 0.939 ±0.11      | 0.298 ±0.04      |
>
> While Online EWC does a very good job on the default 7x7 grid, we can see a downward trend in performance as the number of agents increases. We expect larger grids, longer sequences, and more agents to pose a more complex CL challenge.
>
> **W3 Forced Coordination**
>
> While we already tried a somewhat similar setting in Appendix H, we think this is a great idea. We added a `forced_cooperation` flag to the layout generator, the enabling of which triggers the layout validator to additionally ensure that (1) no walkable path exists between any pair of agents, and (2) for every agent, at least one of the three cook-deliver cycle components (onion <--> pot, pot <--> plate, pot <--> delivery) is unreachable in that agent’s reachable region, while the union over all agents’ reachable regions still admits a valid cycle. In this setting the max soup estimator now finds the best cook–deliver cycle within the union of all agents’ reachable regions. We tested this setting using Online EWC on 20-task sequences with 2-agents.
>
> | Forced Cooperation          | Level 1                 | Level 2                 | Level 3                 |
> |------------------|--------------------------|--------------------------|--------------------------|
> | Off       | 1.612 ± 0.03         | 1.337 ± 0.03         | 1.128 ± 0.11         |
> | On | 1.593 ± 0.13             | 1.167 ± 0.05             | 0.824 ± 0.21             |
>
> The performance gap widens with higher levels. From the training curves and gameplay recordings we noticed that there are more layouts where agents failed to find any solution.

---

> ### Author Response · Authors · 2025-11-20
>
> **W4 Overcooked Not Explained**
>
> >The paper is well-written and clear for the most part, but I think that in section 4.1, the authors should add 1-2 sentences of explanation of the goal of the overcooked environment, and what is the meaning of onions, plates, deliveries, etc. Alternatively, authors could place this explanation in the appendix and refer to it; however, either way, the reader's understanding of the basic environment dynamics should not require them to look beyond the reviewed paper.
>
> We agree that this would be useful. Section 4 currently begins with
> > We present MEAL, the first CMARL benchmark, built on the JaxMARL Rutherford et al. (2024) version of Overcooked.
>
> We shall include the following as a continuation:
> "*The goal in Overcooked is for agents to cooperatively prepare and deliver soup. They must collect onions, place them into pots, wait for the soup to cook, plate the dish, and deliver it to a serving station.*"
>
> **W5 Multi-Output Heads**
>
> > The fact that the main experiments use multi-headed architectures (one head for each task) is mentioned for the first time in section 5.2 - Ablation Study. I think that this should be mentioned earlier (section 5.1 perhaps?), as it is an important (although natural) choice.
>
> This is a fair point. In Section 5 we currently write
> >In our experiments, we adopt the task-incremental continual learning paradigm, in which the task identity is known during both training and evaluation.
>
> We will follow this up with
> > Consequently, we employ a multi-headed architecture, assigning a dedicated output head to each task. This is a common choice for a simplified evaluation setting to better analyze existing methods [1, 2].
>
> Thanks for mentioning this. It was indeed not evident.
>
> **W6 100-Task Experiments**
>
> > In the abstract, the authors claim:
> “MEAL leverages JAX for GPU acceleration, enabling continual learning across sequences of 100 tasks on a standard desktop PC in a few hours.”
> However, most tasks used in the paper are 20. There should be at least one experiment in which authors push the benchmark to the limits and use 100 tasks, which they claim is possible.
>
> First of all, we have made one correction to the abstract since submission. The original claim stated "*a standard desktop PC*", however, this is not accurate, since we used more powerful hardware. We have modified this claim to:
> > enabling continual learning across sequences of 100 tasks on a **single GPU** in a few hours.”
>
> Addressing the more important part of the comment: the submitted version did already include one experiment spanning 100 tasks: the network plasticity evaluation. However, this was without continual evaluation, since we didn't care about the core 3 metrics for that analysis. Since continually evaluating on all tasks in the sequence adds notable overhead, we agree with the reviewer that we haven't provided a proper demonstration.
>
> Since the time of submission, we have included an important experiment to motivate the use of long sequences. Namely, we compare Online EWC and EWC across 100 tasks for all three difficulty levels. We currently only used 3 seeds instead of our default 5. We will extend this to 5 for the camera-ready.
>
> | Level | Method       | Average Score (↑)        | Forgetting (↓)           |
> |-------|--------------|--------------------------|--------------------------|
> | 1     | EWC          | 0.345 ± 0.11             | 0.102 ± 0.07             |
> | 1     | Online EWC   | 1.452 ± 0.05             | 0.010 ± 0.00             |
> | 2     | EWC          | 0.234 ± 0.03             | 0.052 ± 0.04             |
> | 2     | Online EWC   | 0.980 ± 0.03             | 0.056 ± 0.01             |
> | 3     | EWC          | 0.140 ± 0.02             | 0.016 ± 0.02             |
> | 3     | Online EWC   | 0.656 ± 0.08             | 0.093 ± 0.02             |
>
> On shorter sequences of 20 tasks, the two methods behave similarly (Figure 3, top), with standard EWC even slightly outperforming the online variant. When the sequence spans 100 tasks, however, the underlying differences between the methods become clearly visible. We will omit the reasoning behind the diverging behaviour over long sequences of these methods here, since it's not directly relevant to the reviewer's comment. What is relevant, is that results as such really strengthen the need for using long task sequences, which is one of our core motivations for MEAL.
>
>
> [1] Wołczyk, Maciej, et al. "Continual world: A robotic benchmark for continual reinforcement learning." Advances in Neural Information Processing Systems 34 (2021): 28496-28510.
>
> [2] Tomilin, Tristan, et al. "Coom: A game benchmark for continual reinforcement learning." Advances in Neural Information Processing Systems 36 (2023): 67794-67832.

---

> ### Author Response · Authors · 2025-11-20
>
> **W7 Hyperparameters**
>
> > In Appendix B2, the hyperparameter for parallel envs is only 16. This seems peculiar, as the primary speedup from using JAX is the parallelization across a number of environments. In similar works that use JAX for fast benchmarks, the authors typically use environments in the order of thousands or tens of thousands (for example, in the JaxMarl blog post, the authors compare 100, 1k, and 10k environments). What is the reason for such a small number? Have the authors tested larger numbers to achieve higher GPU utilization, and consequently greater learning speed?
>
> Thank you for pointing this out. This is an oversight by us. The values reported in Appendix B2 were from our preliminary runs. While building the initial continual learning pipeline, we used the parameters from JaxMARL for testing, later we adjusted these to fully utilize our hardware. For our experiments in the paper, we used the updated set:
>
> | Hyperparameter            | Value                                 |
> |---------------------------|---------------------------------------|
> | Env. steps per task (Δ)   | 1e8                                   |
> | Parallel environments     | 2048                                  |
> | Rollout length (T)        | 400                                   |
> | Effective batch size      | 2048 × 400 = 819,200                  |
> | Updates per task          | ⌊1e8 / 819,200⌋ = 122                 |
> | Update epochs             | 8                                     |
> | Minibatches per update    | 16                                    |
> | Gradient steps per task   | 122 × 8 × 16 = 15,616                 |
>
> We will update Appendix B2 accordingly. In terms of walltime, a 10-task (1B steps) Level 1 sequence run with EWC, 2 agents, and continual evaluation requires 17 min on average, while a 100-task (10B steps) sequence runs under the same setting for 4 h 36 min on average.
>
>
> **Q1 Scaling Networks**
>
> > Although I don’t think it is necessary in this work, I would like to see a study of how scaling of the network sizes impacts performance. In the current version, the authors use 2x128 MLP networks, which are small even for RL algorithms. Especially in light of recent scaling results (for example, https://arxiv.org/abs/2405.16158, https://arxiv.org/abs/2410.09754), I think it is important to leverage the speed of JAX to provide some indication of whether scaling the networks achieves any performance gain.
>
> Thanks for the suggestion! We tried one extra setting and indeed the performance improved quite substantially. We trained EWC on 20-task sequences. We shall investigate further to find to what extent we can scale the network while still getting performance gains.
>
> | Network           | Level 1                     | Level 2                     | Level 3                     |
> |-------------------|-----------------------------|-----------------------------|-----------------------------|
> | [128,128]  | 1.022 (± 0.18)             | 0.984 (± 0.11)             | 0.622 (± 0.17)             |
> | [256,256,256]   | **1.497** (± 0.07)         | **1.282** (± 0.07)         | **1.011** (± 0.07)         |

---

> > ### Comment · Reviewer_tMtN · 2025-11-22
> >
> > I'd like to thank the authors for engaging with my concerns. I'm raising the score to 6.

---

> > > ### Author Response · Authors · 2025-11-22
> > >
> > > Thank you for considering our rebuttal and for raising your score. May we ask what you still find lacking for a score of 8, and why you consider our work to be only marginally above the acceptance threshold?

---

### Official Review · Reviewer_JNx1 · 2025-10-30

**Soundness:** 3
**Presentation:** 3
**Contribution:** 2
**Rating:** 4
**Confidence:** 4

**Summary:**

The paper introduces MEAL (Multi-agent Environments for Adaptive Learning), a benchmark for continual cooperative multi-agent reinforcement learning (CMARL). MEAL builds upon the Overcooked environment and leverages JAX for GPU-accelerated simulations, enabling scalable training across sequences of up to 100 tasks. The benchmark offers procedural environment generation, controllable difficulty levels, and supports evaluation of both full and partial observability. Six continual learning (CL) baselines (EWC, Online EWC, L2, MAS, AGEM, FT) are implemented and systematically compared. The authors show that while classical CL methods can retain performance in simple settings, they struggle as environment complexity and coordination demands increase.

**Strengths:**

- Timely and relevant benchmark: Addresses the underexplored intersection of continual learning and cooperative multi-agent reinforcement learning, filling a clear gap in existing benchmarks.
- GPU-accelerated simulation: The use of JAX for end-to-end GPU-based training is a notable practical advancement, drastically reducing wall-clock training time and enabling long task sequences (up to 100 tasks on a single GPU).
- Comprehensive baseline coverage: Six continual learning baselines implemented in JAX provide a solid foundation for reproducible comparisons and future benchmarking.
- Clear and organized structure: The paper is well-written and systematically walks through environment generation, difficulty scaling, and evaluation metrics.
- Strong empirical evaluation: Provides clear analyses of forgetting, forward transfer, and average performance across difficulty levels, and meaningful visualizations (Figures 3–5).
- Procedural task generation: Ensures a potentially infinite task space with controlled difficulty, supporting reproducibility through seeded generation.
- Thoughtful design of variants: Inclusion of curriculum and repetition settings
- Insightful ablations: The analysis of multi-head architectures and plasticity loss (Figures 6 and 8) provides valuable insights into continual adaptation in MARL.
- Technical rigor and reproducibility: The inclusion of environment generation algorithms (Appendix A) and ablation studies (e.g., Figure 6) shows commendable transparency and attention to implementation detail.
- Code is provided and contains relevant details, including scripts to reproduce the plots.

**Weaknesses:**

- Claim of first CMARL benchmark requires clarification: The authors state (line 039) that MEAL is the first continual MARL benchmark, but works such as “Multi-Agent Continual Coordination via Progressive Task Contextualization” (Yuan et al., 2024) already explore continual multi-agent coordination.
- Task ID dependence (line 305 & Figure 6): Since the ablation suggests task IDs have negligible effect, it would strengthen the benchmark if baseline results without task ID access were also reported for completeness and fairness.
- Limited task diversity: The benchmark is confined to a single Overcooked cooperative domain with discrete action spaces, which contrasts with more general-purpose benchmarks (e.g., Melting Pot, Continual World) that support broader environmental and control variations.
- Accessibility for new readers: Line 047: The paper assumes prior familiarity with Overcooked; a brief intuitive explanation of its mechanics and why layout changes matter would improve clarity.
- Lines 080–090: Acronyms such as EWC, MAS, and PackNet appear without definition in the main text; a short reminder or a glossary table would help readers less familiar with continual learning.

**Questions:**

- Clarification on novelty claim: Please explicitly discuss how MEAL differs from Yuan et al. (2024) and other CMARL-like studies that included continual coordination experiments.
- Baselines without task IDs: Since Figure 6 shows the negligible effect of task IDs, a results table for “no-task-ID” settings would provide valuable insight into real continual adaptability.
- Explanation of Overcooked: Add a short introductory paragraph early in the paper (around line 047) explaining what Overcooked is and why its layout variability makes it suitable for continual learning.
- Define acronyms early: Ensure EWC, MAS, PackNet, etc., are defined upon first use (lines 080–090).
Per-task evaluation diversity: Why is EWC the only baseline visualized in per-task evaluation (line 297)? Showing others could reveal different forgetting or transfer dynamics.

---

> ### Author Response · Authors · 2025-11-20
>
> **W1 First CMARL Benchmark**
>
> >Claim of first CMARL benchmark requires clarification: The authors state (line 039) that MEAL is the first continual MARL benchmark, but works such as “Multi-Agent Continual Coordination via Progressive Task Contextualization” (Yuan et al., 2024) already explore continual multi-agent coordination.
>
> The "**required clarification**" is as follows: the paper you referred to is not a benchmark, MEAL is a benchmark. We never claimed that we are the first to "*explore continual multi-agent coordination*". Note also that on line 031 we already cite this paper.
>
> **W2 Task ID Ablations**
>
> >Task ID dependence (line 305 & Figure 6): Since the ablation suggests task IDs have negligible effect, it would strengthen the benchmark if baseline results without task ID access were also reported for completeness and fairness.
>
> The brown-colored bars in Figure 6, accompanied by the label "**No Task Id**", already report these results. As the ablation name suggests, these experiments were conducted with *no task ID* for ***completeness*** and ***fairness***.
>
> **W3 Limited Task Diversity**
>
> > Limited task diversity: The benchmark is confined to a single Overcooked cooperative domain [...], which contrasts with more general-purpose benchmarks (e.g., Melting Pot, Continual World) that support broader environmental and control variations.
>
> Indeed. And Continual World is confined to a single robotic-arm manipulation domain, while Melting Pot is confied to a single social-dilemma gridworld domain. Please enlighten us on how these other toy-problem benchmarks you bring as examples are "*more general-purpose*" or "*support broader environmental and control variations*" than MEAL. Note also that in response to reviewer n4Rh, we extensively explain the utility and relevance of MEAL with regard to limited task diversity.
>
> > with discrete action spaces
>
> We are very curious where this allergy to discrete action spaces originates from. Discrete control is not some niche corner case. Many real-world problems are also discrete in nature. Examples include: inventory management [1], healthcare [2], traffic control [3], power management [4], and trading [5]. Not to mention the very recent RL simulation environments [6, 7, 8, 9] that incorporate this.
>
> [1] Mao, Hongzi, et al. "Resource management with deep reinforcement learning." Proceedings of the 15th ACM workshop on hot topics in networks. 2016.
>
> [2] Raghu, Aniruddh, et al. "Deep reinforcement learning for sepsis treatment." arXiv preprint arXiv:1711.09602 (2017).
>
> [3] Wei, Hua, et al. "Colight: Learning network-level cooperation for traffic signal control." Proceedings of the 28th ACM international conference on information and knowledge management. 2019.
>
> [4] Lin, Xue, Yanzhi Wang, and Massoud Pedram. "A reinforcement learning-based power management framework for green computing data centers." 2016 IEEE International Conference on Cloud Engineering (IC2E). IEEE, 2016.
>
> [5] Jiang, Zhengyao, Dixing Xu, and Jinjun Liang. "A deep reinforcement learning framework for the financial portfolio management problem." arXiv preprint arXiv:1706.10059 (2017).
>
> [6] Radji, Waris, Thomas Michel, and Hector Piteau. "Octax: Accelerated CHIP-8 Arcade Environments for Reinforcement Learning in JAX." arXiv preprint arXiv:2510.01764 (2025).
>
> [7] Salaorni, Davide, et al. "Gym4ReaL: A Benchmark Suite for Evaluating Reinforcement Learning in Realistic Domains." Eighteenth European Workshop on Reinforcement Learning.
>
> [8] Ramanujam, Asha, et al. "SafeOR-Gym: A Benchmark Suite for Safe Reinforcement Learning Algorithms on Practical Operations Research Problems." arXiv preprint arXiv:2506.02255 (2025).
>
> [9] Niu, Yazhe, et al. "Lightzero: A unified benchmark for monte carlo tree search in general sequential decision scenarios." Advances in Neural Information Processing Systems 36 (2023): 37594-37635.

---

> ### Author Response · Authors · 2025-11-20
>
> **W4 Overcooked not Introduced**
>
> > Accessibility for new readers: Line 047: The paper assumes prior familiarity with Overcooked; a brief intuitive explanation of its mechanics [...]
>
> This would indeed be useful. Section 4 currently begins with
>
> > We present MEAL, the first CMARL benchmark, built on the JaxMARL Rutherford et al. (2024) version of Overcooked.
>
> We shall include the following as a continuation:
> "*The goal in Overcooked is for agents to cooperatively prepare and deliver soup. They must collect onions, place them into pots, wait for the soup to cook, plate the dish, and deliver it to a serving station.*"
>
> Note that Section 4.1 already describes the core mechanics (movement, interaction rules, cooking dynamics, reward structure) at a sufficient level of detail. If the reader needs a deeper introduction, they can refer to the original work, which we cite.
>
> > [...] and why layout changes matter [...]
>
> In lines 48-52 of the paper we already explain why layout changes matter and justify our choice of overcooked for continual learning:
> > Prior work has shown that agents tend to exploit spurious correlations in fixed layouts, resulting in poor generalization even under minor modifications Knott et al. (2021). This makes Overcooked particularly well-suited for learning continually: even minor layout variations can present a significant challenge.
>
> Later, our experiments empirically validate this point: under the single-output-head setting, agents catastrophically forget learned tasks when faced with slight changes in the kitchen layout.
>
> **W5 Acronyms not Introduced**
>
> > Lines 080–090: Acronyms such as EWC, MAS, and PackNet appear without definition in the main text; a short reminder or a glossary table would help readers less familiar with continual learning.
>
> We don't really see how this is an issue since we provide all the references for these CL methods. If the reader does not know what EWC is, they also won't know what Elastic Weight Consolidation means without opening the relevant paper. It is standard practice in literature to refer to methods by their acronyms, provided that the citation follows.
>
> **Questions**.
> Since all 4 questions effectively restate the weaknesses in different wording, we cannot find anything new to respond to.

---

> > ### Comment · Reviewer_JNx1 · 2025-11-25
> >
> > **W1 First CMARL Benchmark & W2 Task ID Ablations**
> >
> > I thank the authors for these clarifications. The distinction between Yuan et al. (2024) as a method paper versus MEAL as a benchmark is now clear. I acknowledge that the "No Task Id" ablation was already present in Figure 6—this was an oversight on my part.
> >
> > **W3 Limited Task Diversity**
> >
> > I appreciate the authors' detailed response, but I encourage a more constructive framing. Limitations are not criticisms meant to diminish the work—they help the research community understand the scope and applicability of a benchmark.
> >
> > The authors correctly note that Continual World is confined to robotic-arm manipulation and Melting Pot to social-dilemma gridworlds. However, acknowledging MEAL's specific strengths (credit assignment challenges, memory requirements, cooperative coordination) alongside its constraints (single domain, discrete actions, procedural variations rather than fundamentally different tasks) would strengthen the manuscript. This positioning—currently relegated to Appendix L—deserves prominence in the main text to help readers situate MEAL within the broader benchmark landscape.
> >
> > I note that Reviewer n4Rh raised similar concerns about task diversity, questioning whether layout changes alone constitute "truly continual RL." Reviewer gMyk also noted that task diversity is significant beyond mere quantity. A clearer discussion of what types of continual learning challenges MEAL is designed to probe (and which it is not) would address these shared concerns across reviewers.
> >
> > **W4 Overcooked Not Introduced**
> >
> > I appreciate the authors' commitment to add a brief description. To clarify: while lines 48–52 explain *why* layout changes matter, this explanation assumes readers already understand the environment dynamics and task objectives. A sentence or two describing the cook-deliver cycle before explaining why layout changes are challenging would significantly improve accessibility for readers unfamiliar with Overcooked.
> >
> > **W5 Acronyms Not Introduced**
> >
> > I respectfully disagree with the authors' position that citations alone suffice. Accessible scientific writing does not require readers to open multiple papers to understand basic terminology. A brief parenthetical—e.g., "EWC (Elastic Weight Consolidation)"—upon first use costs nothing and benefits readers who may be experts in MARL but less familiar with continual learning methods, or vice versa. This is particularly important for a benchmark paper intended to serve a broad audience. The argument that "everyone does this" conflates common practice with best practice.
> >
> > **Q5: Per-Task Evaluation Diversity (Unanswered)**
> >
> > My question about why only EWC is visualized in per-task evaluation (line 297) remains unanswered. Showing additional baselines could reveal different forgetting or transfer dynamics and would strengthen the empirical analysis.
> >
> > ---
> >
> > **Cross-Reviewer Context**
> >
> > Several concerns I raised align with those of other reviewers:
> >
> > - **Task diversity**: Reviewers n4Rh, gMyk, and I all raised questions about the limited scope of procedural variation. The authors have addressed this by adding new environment dynamics (pot size, soup timer, sticky actions, slippery tiles), which represents meaningful progress.
> > - **Accessibility**: Reviewer tMtN also requested better explanation of Overcooked mechanics, supporting my W4 concern.
> >
> > **Recommendation**
> >
> > The core contribution—a JAX-accelerated, procedurally-generated benchmark for continual MARL—remains valuable. The authors have made substantive additions that strengthen the work. My remaining concerns are primarily about presentation and accessibility rather than technical substance. With revisions to improve clarity and explicit positioning of MEAL's scope, this work would be strengthened considerably.

---

> > > ### Author Response · Authors · 2025-11-25
> > >
> > > We thank the reviewer for the follow-up and for acknowledging that many of the raised points were addressed. Based on your latest message, it appears that **3 concerns remain**: (1) missing acronyms, (2) per-task evaluation plots, and (3) task diversity. We shall address the straightforward ones first.
> > >
> > > **W5 Acronyms (1)**. Since the reviewer insists, we don't mind including the extra 83 keystrokes (+6 for parentheses) to write the full names of EWC, MAS, and AGEM.
> > >
> > > **Q5 Per-task evaluation curves (2)**. These results are already present in Figure 17 of the Appendix for all baselines.
> > >
> > > **W3 Task Diversity (3)**
> > >
> > > > alongside its constraints
> > >
> > > We don't consider the constraints you list as notable constraints nor strong weaknesses:
> > >
> > > > single domain / procedural variations rather than fundamentally different tasks
> > >
> > > First, in response to Reviewer tMtN, we included an extra environment, **JaxNav**, to show that the **MEAL framework is not limited to Overcooked**. Second, MEAL also includes a setting for **continual partner adaption**, which we outline in Section 4.4, and further analyze in Appendix J. This changes the learning problem fundamentally, not merely the map layout. Third, we don't think that having a benchmark in a "*single domain*" with only procedural variants is a weakness to begin with. As we noted, many widely used RL benchmarks focus on a single domain: Continual World, SMAC, ViZDoom, CARLA, while ProcGen also just uses procedural variations. We don't think a benchmark necessarily ought to compose an unbounded array of different environments. As we explain in our response to Reviewer n4Rh, there is much value in analysing a single domain in depth rather than scratching the surface of a broad range of environments.
> > >
> > > > discrete actions
> > >
> > > As we already noted in our rebuttal, discrete-action RL is used extensively in real-world systems: inventory management, treatment planning, traffic control, power management, trading, and many modern RL benchmarks (e.g., OCTaX, SafeOR-Gym, LightZero). Nothing about CL or MARL inherently requires continuous control.
> > >
> > > You point out that reviewers n4Rh and gMyk raised similar concerns about task diversity. Note that we have provided elaborate responses about this concerning (1) introducing new elements of non-stationarity, (2) showing that CL performance of agents collapses even under trivial layout changes, suggesting that this is a complex enough evaluation setting, (3) introducing an entirely new environment, JaxNav, to show MEAL's compatibility. Do you feel that something is missing/lacking in those responses? As it stands, we are slightly confused. You note that your ***remaining concerns are primarily about presentation and accessibility rather than technical substance***. Given that:
> > >
> > > - your initial review listed a whole **10 strengths**,
> > > - your previous comment states that many of your concerns were adequately addressed,
> > > - the outstanding issues now seem to be minor writing adjustments,
> > > - in the *Cross-Reviewer Context* section you praise us for not only addressing some of your concerns but also those of other reviewers,
> > >
> > > yet the score remains a 4. Given all of this, we struggle to understand what separates a score of 4 from a score of 8 in your evaluation, as the gap feels disproportionately large, almost as if the scale behaves more like a logarithmic one than a linear one. At this point, it is unclear to us what additional improvements in presentation or accessibility you feel are necessary for the work to move from a score of 4 to a score of 8 (Accept). If all the reviewer is asking for, is that we move some points of limitations from Appendix L to the main text, then we are happy to do so.

---

> > > > ### Comment · Reviewer_JNx1 · 2025-11-26
> > > >
> > > > I want to note that the discussion phase is not yet over, and directly pushing for score increases may not be the most productive strategy at this stage.
> > > >
> > > > It appears my suggestions to make this benchmark more accessible—by providing clearer information to readers—were dismissed as merely "83 extra keystrokes." I wish to emphasize that making research accessible and easy to follow is of significant benefit to the community. Similarly, pointing to Appendix Figure 17 for per-task evaluation curves, while technically addressing Q5, underscores my broader concern: key information should be discoverable in the main text, not buried in supplementary material.
> > > >
> > > > I can only reiterate that **limitations are not meant to downplay this work**—they help readers and future benchmark users classify which types of challenges this benchmark addresses. Acknowledging that MEAL excels at cooperative coordination challenges, while being focused on a single procedural domain, is a strength of scientific writing, not a weakness of the contribution.
> > > >
> > > > Regarding W3: I acknowledge the authors' additions (JaxNav, partner adaptation, new dynamics). These are valuable. My point remains that explicitly positioning MEAL's scope in the main text—rather than defending against perceived criticism—would benefit readers.
> > > >
> > > > **Accessibility and presentation quality** should emerge from clear, well-structured writing rather than defensive and harsh responses. It is the authors' responsibility to make contributions transparent and easy to accesss.
> > > >
> > > > Finally, I would appreciate if future exchanges remain professional in tone. Phrases questioning my evaluation scale or implying that my concerns are disproportionate do not advance the scientific discussion.
> > > >
> > > > I am willing to raise my score with the expectation that the authors (1) maintain professional communication, (2) incorporate the agreed-upon clarifications (acronym definitions, Overcooked description), and (3) position MEAL's unique strengths and intended scope clearly in the main text. This is not about listing weaknesses—it is about articulating the benchmark's design choices and the specific continual learning challenges it is best suited to evaluate.

---

> > > > > ### Author Response · Authors · 2025-11-26
> > > > >
> > > > > > I would appreciate if future exchanges remain professional in tone
> > > > >
> > > > > We apologize if any part of our earlier responses came across as unprofessional or upset the reviewer. Our intentions were not to dismiss or undermine any feedback, but to understand how we could improve the paper in ways that meaningfully address the reviewer’s concerns.
> > > > >
> > > > > > directly pushing for score increases may not be the most productive strategy at this stage.
> > > > >
> > > > > We would like to clarify that our goal was not to directly push for a score increase. Since we deem the score as a direct indicator of the quality of the paper, we noticed a mismatch between the score and the discussion that was unfolding. This made us confused about whether our responses up to that point were of any merit, and what further improvements we could make to address the presentation and quality of our work. We hence explicitly wished to ask what the reviewer would suggest that we add or change.
> > > > >
> > > > > > It appears my suggestions to make this benchmark more accessible—by providing clearer information to readers were dismissed as merely "83 extra keystrokes.
> > > > >
> > > > > We did not mean this in an ill-intended way. The intent was to emphasize that adding acronym explanations is a simple edit on our side, not to diminish the reviewer’s request. We fully agree that benchmark papers should be accessible to a broad audience and that clarity of presentation is essential.
> > > > >
> > > > > > Similarly, pointing to Appendix Figure 17 for per-task evaluation curves, while technically addressing Q5, underscores my broader concern: key information should be discoverable in the main text, not buried in supplementary material.
> > > > >
> > > > > We are more than happy to include additional figures and details in the main text, however, the ICLR format limited our initial submission to a 9-page limit. We had made our best attempts to include the most essential content in the main paper, and might have missed some key aspects. We thank the reviewer for pointing this out. While we agree that Figure 17 is valuable in showing how the baselines compare to one another, it occupies nearly an entire page. Hence, we kept it in the appendix and only displayed the curves of EWC in the main part. We have revised the paper by providing an explicit reference to Figure 17 and including a brief description to ensure these results are easy to locate and interpret. We very much welcome suggestions for other key points in the appendix that ought to be deemed worthy of being included in the main text.
> > > > >
> > > > > > Accessibility and presentation quality should emerge from clear, well-structured writing rather than defensive and harsh responses. It is the authors' responsibility to make contributions transparent and easy to access. I can only reiterate that limitations are not meant to downplay this work—they help readers and future benchmark users classify which types of challenges this benchmark addresses. Acknowledging that MEAL excels at cooperative coordination challenges, while being focused on a single procedural domain, is a strength of scientific writing, not a weakness of the contribution. [...] My point remains that explicitly positioning MEAL's scope in the main text—rather than defending against perceived criticism—would benefit readers.
> > > > >
> > > > > Thank you for clarifying this point. We have to admit that we misunderstood the prior comments, and we agree that clear positioning and outlining of the benchmark’s scope are important for readers. We have revised Section 4 to state MEAL's scope more explicitly. Second, we have moved the Limitations to the main part of the paper to make the gaps and shortcomings of our work more obvious to the reader.
> > > > >
> > > > > We deeply appreciate the efforts for reviewing our work and making it clear what steps we should take to improve it. We have made the necessary edits to the paper and uploaded a new version. To make the revisions easier to locate, we have temporarily highlighted them with a different color. In particular, we have incorporated (1) a brief introduction to Overcooked, (2) explicit acronym definitions, (3) a positioning of MEAL’s scope and design focus, (4) a cross-reference and brief description of the per-task evaluation curves of all baselines, and (5) the moving of the limitations to the main part of the paper. Please note that the edits might have temporarily messed up the formatting of some figures/tables. We will fix this in the Camera-Ready version. Please let us know if we missed something.

---

> > > > > > ### Comment · Reviewer_JNx1 · 2025-11-27
> > > > > >
> > > > > > I thank the authors for their thoughtful response and for the constructive revisions to the manuscript. The additions represent meaningful improvements to the paper's accessibility.
> > > > > >
> > > > > > I acknowledge that the revised manuscript addresses the core concerns I raised. However, I would like to offer one additional suggestion regarding the framing of limitations:
> > > > > >
> > > > > > **Framing of Limitations**: The current revision reads somewhat as a downplay of the work rather than a balanced positioning. For instance, the statement "First, MEAL is restricted to discrete action spaces, limiting its applicability" is not entirely accurate - as the authors themselves compellingly argued in the rebuttal, discrete-action RL is extensively used in real-world systems (inventory management, treatment planning, traffic control, power management, trading) and many modern benchmarks. I would encourage the authors to frame this section as "Design Choices and Scope" or similar, presenting both the strengths and the boundaries of the benchmark in a balanced manner. This would be more consistent with the strong arguments the authors already made during the discussion.
> > > > > >
> > > > > > Some minor points remain, such as unexplained approaches like IPPO, but I do not wish to make these central to the evaluation, as they pertain more to writing polish than technical substance.
> > > > > >
> > > > > > Given the authors' nuanced and detailed responses throughout this discussion, their willingness to incorporate feedback, and the concrete revisions made to the manuscript, I am willing to increase my score. I expect the authors to continue refining the presentation in the camera-ready version, particularly because of the still remaining page real estate.

---

### Official Review · Reviewer_n4Rh · 2025-10-31

**Soundness:** 2
**Presentation:** 3
**Contribution:** 1
**Rating:** 2
**Confidence:** 3

**Summary:**

This paper proposes MEAL, a new benchmark for continual multi-agent reinforcement learning based on the Overcooked cooperative multi-agent RL environment. MEAL adds procedural generation to Overcooked, a grid world, to vary grid size and obstacle density. MEAL also uses JAX for GPU acceleration, enabling the benchmarking of continual MARL over longer task sequences on a single desktop GPU in a few hours. Experiments by naively combining continual RL methods with standard MARL methods show that approaches fail in more complex continual MARL settings where non-stationarity arises from both multi-agent dynamics and the task distribution.

**Strengths:**

- The problem setting is interesting and novel. I am not aware of any prior work considering the continual multi-agent reinforcement learning paradigm.

- The paper is generally well-written and organized. The experiments use standard continual RL evaluation metrics including forgetting and forward transfer.

**Weaknesses:**

1. MEAL only considers non-stationarity in changing environment layouts (Line1326) while the observation space and action space are the same between "tasks" in MEAL. Given this, I am not certain if the setting studied in this paper is truly continual RL. In particular, this distribution shift from procedural generation is most similar to individual environments in Procgen [1]. For comparison, Jelly Bean World [2] uses an infinite grid world to achieve non-stationarity, Continual World [3] uses different objects in the scene, and CORA [4] uses different tasks within game environments (which have different observation space, action space, and environment dynamics altogether).

2. Table 1 claims that MEAL covers infinite tasks, yet only 3 difficulty levels are studied. Use of procedural generation is not sufficient to claim infinite tasks, given that some of the other benchmarks in the table also use environments with procedural generation such as Procgen.

3. The paper primarily focuses on a multi-head output setting, instead of using a shared output head. Ablation in Figure 6 shows that EWC completely fails on the MEAL benchmark when using a shared output head. Using individual heads while claiming that MEAL spans an infinite number of tasks (Table 1) would result in unbounded memory usage to store additional model parameters. Furthermore, careful evaluation of the shared output head setting may show that other methods (ie. replay-based vs. penalty-based) outperform EWC. CLEAR, a replay-based method, drastically outperforms CLEAR on single-agent continual RL when using a shared output head.

[1] https://arxiv.org/abs/1912.01588

[2] https://arxiv.org/abs/2002.06306

[3] https://arxiv.org/abs/2105.10919

[4] https://arxiv.org/abs/2110.10067

[5] https://arxiv.org/abs/1811.11682

**Questions:**

1. It would be beneficial for MEAL to also evaluate changing environment dynamics, rather than simply environment layout within Overcooked. This may be achieved by adding "kitchen" / "recipe" formats beyond the current onion soup recipe. Another path to consider would be to add multi-agent support for other JAX-based grid world environments such as Craftax [1], Naxvix [2], or XLand-Minigrid [3]. Or to directly incorporate other multi-agent RL environments built in Jax such as MAC [4] or SocialJax.

2. Regarding Table 3, how easy is it to scale MEAL to more than 3 agents? Given the performance boost from Jax, what about training with 10 agents? 100 agents?

[1] https://arxiv.org/abs/2402.16801

[2] https://arxiv.org/abs/2407.19396

[3] https://arxiv.org/abs/2312.12044

[4] https://arxiv.org/abs/2503.14576

[5] https://arxiv.org/abs/2503.14576

---

> ### Author Response · Authors · 2025-11-22
>
> **W1 Non-stationarity**
>
> *MEAL only considers non-stationarity in changing environment layouts (Line1326) while the observation space and action space are the same between "tasks" in MEAL. Given this, I am not certain if the setting studied in this paper is truly continual RL. In particular, this distribution shift from procedural generation is most similar to individual environments in Procgen [1]. For comparison, Jelly Bean World [2] uses an infinite grid world to achieve non-stationarity, Continual World [3] uses different objects in the scene, and CORA [4] uses different tasks within game environments (which have different observation space, action space, and environment dynamics altogether).*
>
> We think the reviewer has raised a valid point but we disagree with the extent of the criticism. We shall first discuss the points of disagreement and then provide a remedy for this issue.
>
> > Given this, I am not certain if the setting studied in this paper is truly continual RL
>
> We are very curious what the reviewer considers as "*truly continual RL*". We don't see how MEAL falls short in the "*trulyness*" compared to these other benchmarks brought as examples.
>
> > this distribution shift from procedural generation is most similar to individual environments in Procgen [1]
>
> And this is exactly why many researchers have opted to use environments in Procgen to study continual learning. Why is this an issue?
>
> >Jelly Bean World [2] uses an infinite grid world to achieve non-stationarity,
>
> Jelly Bean World studies a different formulation of CL. Instead of clearly defined task sequences like in Continual World/COOM/CORA, it has a single infinte-horizon evolving task. We don't see why one CL formulation should be considered more "*true*" than another. It depends on the eventual application/domain.
>
> >Continual World [3] uses different objects in the scene
>
> Yes, and we use different grid layouts. What is more "*truly continual RL*" about *using different objects* in Continual World compared to *using different layouts* in MEAL?
>
> > CORA [4] uses different tasks within game environments (which have different observation space, action space, and environment dynamics altogether).
>
> While CORA includes 4 platforms (Atari, Procgen, MiniHack, and CHORES), it does not compose CL sequences across them. Each sequence remains confined within a single platform, meaning the observation and action spaces are consistent throughout the sequence.
>
> We don't think that achieving truly Continual RL requires making the simulation environment as different and diverse as possible at all costs. Similary, in real life, I doubt you would expect a policy trained to operate a self-driving car to be adapted to fold laundry or wash dishes as its next task. We think it is far more realistic that the observation and action spaces stay constant across the continual learning *life-cycle*. If we start changing obs/action spaces, we introduce structural discontinuities that break shared representations by design. This would also require structural modifications to the model and make “*continual*” learning close to multi-domain adaptation instead. Consider, for instance, an assembly robot, that perceives the world through a camera and its arm joint locations. It is more natural to view continual learning as the robot being tasked with assembling new objects over time, rather than periodically altering its embodiment, such as attaching a second camera or an extra arm on one day and removing it the next.
>
> From another perspective, rather than seeking diversity for its own sake at all costs, we should instead ask what challenges current CL methods. Relying on well-established metrics, our results already demonstrate that even seemingly simple layout changes create notable difficulty. For instance, on hard layouts with three agents, the average score is as low as 0.117, and using single-output heads further diminishes scores to 0.05 (the agent is unable to remember more than 1/20 tasks). In other words, MEAL already exposes significant limitations of current methods. We don't see how adding further layers of difficulty or non-stationarity would make the setting “*more truly continual*”.
>
> **(continued in the next comment)**

---

> ### Author Response · Authors · 2025-11-22
>
> With that being said, we believe that MEAL already constitutes a valid continual RL benchmark in its current form. Nonetheless, we agree that incorporating additional sources of non-stationarity beyond layout changes could further enrich the environment, offering new evaluation settings and further opportunities to study the behavior of CL methods. To this end, we introduce 4 factors of non-stationarity:
> 1. **Pot Size**. Currently, each soup requires 3 onions. We randomize this value between 1-5 for each pot when starting a new task.
> 2. **Soup Timer**. Currently, each pot takes 20 game tics to cook the soup. We randomize this value between 10-30 for each pot when starting a new task.
> 3. **Sticky Actions**. Inspired by the Atari Learning Environment (ALE) [1], we introduce a probability that an agent repeats its previous action. This probability scales with difficulty: 10% (Level 1), 20% (Level 2), and 30% (Level 3).
> 4. **Slippery Tiles**. In each task, 25% of the grid's walkable tiles are *slippery*. When stepping on these tiles, there is a probability that the agent will *slip* and move to a random adjacent tile in the next step. This probability scales with difficulty: 35% (Level 1), 50% (Level 2), and 65% (Level 3).
>
> We probe each factor individually to see how it affects performance and also test the effect of combinining all. We train Online EWC with each on 20-task sequences.
>
> |Dynamics|Level 1|Level 2|Level 3|
> |---|---|---|---|
> |Default|1.612 ± 0.03|1.337 ± 0.03|1.128 ± 0.11|
> |Pot Size|1.588 ± 0.08|1.284 ± 0.04|1.017 ± 0.05|
> |Soup Timer|1.547 ± 0.11|1.269 ± 0.03|1.060 ± 0.05|
> |Sticky Actions|1.475 ± 0.03|1.177 ± 0.03|0.839 ± 0.11|
> |Slippery Tiles|1.365 ± 0.03|1.161 ± 0.02|0.832 ± 0.05|
> |**Combined**|1.124 ± 0.07|0.948 ± 0.03|0.666 ± 0.04|
>
> Individually, each source of non-stationarity mildly reduces performance. When combined, their effects accumulate and substantially increase task difficulty.
>
> **W2 Infinite Tasks**
> *Table 1 claims that MEAL covers infinite tasks, yet only 3 difficulty levels are studied. Use of procedural generation is not sufficient to claim infinite tasks, given that some of the other benchmarks in the table also use environments with procedural generation such as Procgen.*
>
> > yet only 3 difficulty levels are studied
>
> We’re unsure what the reviewer expects from “*studying higher levels*”. Each difficulty level increases the grid dimensions by 2 (e.g., Level 4: 13×13, Level 5: 15×15, …), and the number of possible layouts grows combinatorially. Beyond the three levels already examined, the diversity is effectively unbounded, adding more would only inflate computation, not insight. We already showed that all baselines perform poorly on Level 3, so we didn't see a need to evaluate higher levels.
>
> > Use of procedural generation is not sufficient to claim infinite tasks
>
> We adopted the *tasks* terminology, as it is common in task-incremental CRL literature, where a sequence is composed of a series of tasks. To avoid confusion we shall modify the main text to claim that MEAL provides **infinite layouts**.
>
> Second, to better reflect what matters for CL, we shall also revise Table 3 to report the max sequence length supported by each benchmark (i.e., the longest out-of-the-box task sequence), instead of the raw number of tasks. For non-CL benchmarks, this will be listed as N/A.
>
> > some of the other benchmarks in the table also use environments with procedural generation such as Procgen
>
> MPE environments are hand-crafted and fixed. Google Football offers fixed scenarios and levels with minor randomization but no PCG. All JaxMARL envs rely on predefined scenarios, with randomness limited to starting locations and opponent behaviour. Meltingpot only randomizes object placements, agent roles, spawn and respawn positions. All COOM tasks have fixed layouts, with random enemy behaviour and item spawns. CORA composes 4 task sequences of Atari (6), Procgen (6), MiniHack (15) and CHORES (4) envs. Only ProcGen indeed uses PCG.
>
>
> [1] Machado, Marlos C., et al. "Revisiting the arcade learning environment: Evaluation protocols and open problems for general agents." Journal of Artificial Intelligence Research 61 (2018): 523-562.

---

> ### Author Response · Authors · 2025-11-22
>
> **W3 Multi-Output Heads**
>
> >The paper primarily focuses on a multi-head output setting, instead of using a shared output head.
>
> In our view, the goal of an RL benchmark is to both (1) provide a significant (and preferably scalable) challenge (2) while offering a setting which is solvable by existing methods to an extent that allows grounds for analysis and comparisons. To facilitate the latter, we therefore intentionally opted for the easier multi-output-head variant as the default evaluation setting for our experiments. Note that Continual World [1] and COOM [2] also adopt this as their default evaluation setting. To support the former goal of providing a meaningful challenge, the poor performance under the shared-output-head setting shows the degree of potential MEAL has in posing a challenge to more capable methods in the time to come.
>
> >Ablation in Figure 6 shows that EWC completely fails on the MEAL benchmark when using a shared output head.
>
> As you correctly pointed out, EWC and L2 completely fail with a single output head. If we had used this setting for all our analysis (partial obs, curriculum learning, partner adaptation, difficulty levels, different number of agents, etc.), we would not have been able to draw very meaningful conclusions. Also, if we had only reported 0 performance for all baselines, it would raise the question whether MEAL sequences are solvable at all. Hence we opted for the multi-head variant for our core analysis.
>
> >Using individual heads while claiming that MEAL spans an infinite number of tasks (Table 1) would result in unbounded memory usage to store additional model parameters.
>
> The fact that MEAL can generate an unbounded number of layouts is separate from the question of how the CL algorithm's policy is parameterized. We never claimed that multi-head networks are the mandatory or canonical way to interact with MEAL.
>
> The real weakness on our side is that we were not sufficiently explicit about the intended evaluation protocol. We will revise Section 4.4 to clarify that the benchmark’s expected evaluation mode is "*proper CL*" without access to privileged information such as the task ID, which naturally rules out multi-head policies. Section 5 will then make clear that multi-head variants are used in our experiments for better understanding baseline behavior.
>
> **Q1 Environment Dynamics**
>
> > It would be beneficial for MEAL to also evaluate changing environment dynamics, rather than simply environment layout within Overcooked. This may be achieved by adding "kitchen" / "recipe" formats beyond the current onion soup recipe. Another path to consider would be to add multi-agent support for other JAX-based grid world environments such as Craftax [1], Naxvix [2], or XLand-Minigrid [3]. Or to directly incorporate other multi-agent RL environments built in Jax such as MAC [4] or SocialJax.
>
> We discussed the new enviorment dynamics under Weakness 1. As for the suggestion to extend MEAL with additional environments such as Craftax or Naxvix, we believe there is more value in maintaining a focused and coherent benchmark design than pure quantity. Unlike frameworks such as SocialJax, MeltingPot or JaxMARL, which provide a large collection of distinct games, MEAL emphasizes ***depth***. It deliberately focuses on a single environment, that we study thoroughly across failure modes, partner adaptation, curriculum learning, network plasticity, difficulty levels, partial obs, different number of agents etc. Introducing extra environments would dilute this focus and make our appendix become an entire book if we were to study all of them to the same extent as we studied MEAL envs. Last, we are unsure what MAC is, since both references [4] and [5] appear to point to SocialJax. Is the reviewer referring to SMAX?
>
> **Q2 Scaling Agents**
>
> >Regarding Table 3, how easy is it to scale MEAL to more than 3 agents? Given the performance boost from Jax, what about training with 10 agents? 100 agents?
>
> We tried a few runs on a 20-task sequence using EWC with 10, 25, and 100 agents. Running with 100 agents required enlarging the grid to 13×13, as the hard layout could not physically accommodate all agents. The average score with 10 agents was 0.06, as the environment became too crowded for navigating, and with 25 and 100 agents the score unsurprisingly dropped to 0.00. Below we show the average training time of a single task in the sequence.
>
> |Agents|Time (mm:ss)|Relative Increase|
> |:---:|:----------:|:----------------:|
> |1|1:27|—|
> |10|2:44|+88%|
> |25|5:25|+274%|
> |100|31:01|+2040%|
>
> Although we did not encounter this issue in our experiments, it should be noted that increasing the number of agents proportionally raises GPU memory requirements during JIT compilation. If the computational graph no longer fits on the GPU, one would need to reduce either the number of parallel environments or the rollout length, either of which would increase total training time.

---

### Official Review · Reviewer_gMyk · 2025-10-31

**Soundness:** 3
**Presentation:** 4
**Contribution:** 2
**Rating:** 4
**Confidence:** 3

**Summary:**

MEAL is a benchmark for continual multi-agent reinforcement learning. This suite of JAX environments evaluates continual learning methods in 100 tasks and demonstrates that simple combination of MARL and CL, struggle to complex settings.

**Strengths:**

There is a growing need for increasingly numerous tasks in CRL and Multi-task RL, which this benchmark addresses. The presentation of results and discussions are strong and clear, with useful ablations and performance measures (forgetting, dormant ratio, etc). The paper proposes numerous interesting appendices and adjacent settings such as curriculum learning are a nice addition.

**Weaknesses:**

The claim it is the first continual RL library to leverage JAX should carefully distinguish itself from existing, non-benchmark libraries, such as ReDo (https://github.com/google/dopamine/tree/master/dopamine/labs/redo). Although PyTorch based, Plasticine is also a relevant comparison (https://github.com/RLE-Foundation/Plasticine/tree/main). Whilst there are clearly defined contributions in procedural generation and establishing CL, POMDP and Curriculum settings, the framework seems to also re-implement a lot of functionality from the Overcooked Carrol et al. (2019) paper which limits it's novelty. The claim that MEAL has infinite tasks, whilst true, is somewhat misleading as the diversity between tasks is also of significance. I.e the tasks in Continual World are very distinct, which is perhaps why algorithms trained on MEAL are able to perform well even after numerous task changes. The relatively minor increment in dormancy and plasticity-loss makes it harder to evaluate CL methods against each other.

Similarly using different output heads for different tasks means a portion of the parameters aren't experiencing any distribution shifts, which is why many works omit this privileged information. Whilst the proposed baselines are relevant (albeit slightly dated), it would be interesting to see an optimization method (CBP, ReDo, or related).

**Questions:**

Are you able to elaborate on the intuition as to why harder tasks cause greater plasticity loss? Is there greater shifts in task distribution?

---

> ### Author Response · Authors · 2025-11-14
>
> > The claim it is the first continual RL library to leverage JAX should carefully distinguish itself from existing, non-benchmark libraries
>
> Thank you for pointing this out. Indeed, we have mistakenly written in the introduction that “*MEAL is also the first continual RL **library***”. To clarify, we have no intention of claiming this, simply because MEAL is not an RL library. While RL libraries, such as Dopamine, aim to provide (JAX-based) implementations of (Continual) RL algorithms, MEAL, on the contrary, is a benchmark. While we do implement several CL methods in JAX adapted for RL, this is not the core focus. In MEAL, the entire continual learning pipeline is end-to-end written in JAX, including the environment. We will correct this segment in the paper to "*MEAL is also the first continual RL **benchmark***".
>
> >The claim that MEAL has infinite tasks, whilst true, is somewhat misleading as the diversity between tasks is also of significance.
>
> Would the reviewer be happy if we changed this claim to "*MEAL has infinite **layouts***" instead? We will then further clarify that when we refer to tasks (as common in CRL literature), we mean to refer to a change of layout (except for the partner adaptation setting).
>
> > I.e the tasks in Continual World are very distinct, which is perhaps why algorithms trained on MEAL are able to perform well even after numerous task changes.
>
> With the example of Continual World [1], indeed, conceptually the difference between using a robot arm to place a cube on top of another and opening a drawer is greater, compared to taking 3 steps left or 2 steps right to locate an onion pile in MEAL. Although these small structural variations seem very subtle, we empirically show they yield surprisingly strong learning challenges. For instance, while EWC achieves 0.60 average performance on CW20 in Continual World, it only obtains 0.178 on Level 3 in MEAL with 20-task sequences. Note that MEAL enables trivially scaling up difficulty in an unbounded manner, while in Continual World, it is neither trivial nor evident how to do this. We therefore don't fully agree that "*algorithms trained on MEAL are able to perform well*", as this is not the case for higher levels.
>
> > The relatively minor increment in dormancy and plasticity-loss makes it harder to evaluate CL methods against each other.
>
> As we state in Section 4.5, the core evaluation metrics of MEAL are (1) Average Performance, (2) Forgetting, and (3) Forward Transfer, following prior CRL benchmarks that equally deem these as valuable metrics [1, 2]. Our results under these metrics show a clear performance difference across the baselines, as seen in Figure 3 (top) and Table 2, so we don't see an issue to "*evaluate CL methods against each other*" on MEAL. Our neuron dormancy analysis was included primarily to demonstrate MEAL’s capacity for long-term continual evaluation, not as a core benchmark metric. Few works have investigated plasticity loss across 1 billion time steps. More importantly, we only report dormancy for Fine-Tuning, so there is no telling what plasticity metrics other baselines would yield and how minor or major the increment might be.
>
> > Similarly using different output heads for different tasks means a portion of the parameters aren't experiencing any distribution shifts, which is why many works omit this privileged information.
>
> Our aim is to propose a benchmark, not a new continual learning method. We intentionally opted for the easier multi-output-head variant as the default evaluation setting for our experiments, to be better able to analyze and compare existing methods. As we show in our ablation study, the performance of baselines drops to near zero with single output heads. Using this as a default setting for existing baselines would not be very meaningful, as we couldn't draw any meaningful comparisons. On the contrary, if we were to report 0 performance for all baselines and nothing else, it would raise the question whether MEAL sequences are solvable at all. Of course, we fully agree that for true continual learning, this privileged information should be hidden (which can be configured by the user via switching a single CLI flag). We would even claim that the low performance in the more realistic single output head setting only shows how much potential MEAL has in posing a challenge to more advanced methods in the time to come. Also, please note that Continual World [1] and COOM [2] also adopt this as their default evaluation setting. Continual World states: "*We rely on using a separate head for each new task, similar to many works on continual learning*".
>
> [1] Wołczyk, Maciej, et al. "Continual world: A robotic benchmark for continual reinforcement learning." Advances in Neural Information Processing Systems 34 (2021): 28496-28510.
>
> [2] Tomilin, Tristan, et al. "Coom: A game benchmark for continual reinforcement learning." Advances in Neural Information Processing Systems 36 (2023): 67794-67832.

---

> > ### Author Response · Authors · 2025-11-14
> >
> > > *Are you able to elaborate on the intuition as to why harder tasks cause greater plasticity loss? Is there greater shifts in task distribution?*
> >
> > We do not wish to claim that harder layouts cause greater **plasticity loss**, simply because we never empirically show this. We only show that longer sequences (repetitions of sequences) cause this. We don't compare this across difficulty levels. Although transfer and plasticity can be conceptually related, and we do show that higher levels yield lower forward transfer, we have not empirically verified whether the same is true for plasticity.
> >
> > On the intuition side of things, you are very correct that MEAL layouts of higher difficulty indeed correspond to larger shifts in the underlying task distribution. The grids become larger and the number of walls (obstacles) increases. The agents need to take more deliberate steps to complete the recipe. As a result, it is hence less likely that these long learned patterns can be transferred to new layouts. This is the intuition behind why forward transfer suffers. If we were to extend this reasoning to plasticity, we can hypothesize that the shared trunk of the network must undergo deeper rewiring to acquire new behaviors. Such shifts accumulate and have the potential to progressively saturate network capacity, leading to reduced adaptability and higher dormancy.
> >
> > In Appendix F, we mistakenly claim that "Higher level layouts also add demands for plasticity and transfer." We will correct this to "Higher level layouts also add demands for transfer."
> >
> > We hope these clarifications resolve the reviewer’s concerns and clarify our intended claims. Please let us know if any aspect remains unclear or requires further elaboration.

---

> ### Author Response · Authors · 2025-11-25
>
> Dear Reviewer,
>
> We hope you are doing well. Since it has been 9 days since we submitted our rebuttal, we wanted to kindly check in. We would be grateful if you could find a moment to review our responses, as it would still give us time to address any remaining concerns before the rebuttal period ends.
>
> Thank you for your time and consideration.

---

### Author Response · Authors · 2025-11-28

Dear Reviewers **gmYk**, **n4Rh**, and **tMtN**,

As the rebuttal period is approaching its end in a few days, we would like to kindly follow up. We have tried to carefully address all points raised so far, and we would greatly appreciate it if you could briefly indicate:

1. Which of your original concerns you consider addressed by our rebuttal and revisions,
2. Which concerns you feel remain insufficiently addressed, and why,
3. What score would you consider given our revisions and rebuttal, and
4. What further additions/clarifications would you expect from us to justify a higher score.

This would help us understand how our current responses are being received and what concrete steps we can still take within the remaining rebuttal window (and, if needed, for a future revision of the work).

Thank you again for your time and effort in reviewing our submission.

---

### Author Response · Authors · 2025-11-29
**Rebuttal Summary**

Dear AC,

In light of the recent changes to the review process, we would like to summarize the core concerns raised by each reviewer and how we addressed them in our rebuttal. We shall go through each reviewer individually and categorize their points by:
1. Concerns that revealed a real gap and for which we implemented new features, ran more experiments, or made substantial revisions to the paper. These are the issues that meaningfully improved the work and/or required non-trivial additional effort.
2. Minor or clarification-only issues that did not require any code additions or new experiments, and were resolved through brief explanations, small corrections, or addressing misunderstandings.

Our goal here to provide an overview of how the rebuttal process went from our point of view.

## Reviewer gMyk

### Substantial concerns
1. Claiming **infinite tasks** is an overstatement because we only use PCG for layouts.

On the one hand, we use the **task-incremental learning** CL formulation, where a task is defined by a **distribution shift**. In principle, PCG layouts satisfy this. Also, we believe this aligns better with prior CL literature. On the other hand, this might indeed sound like an overstatement compared to other works. We proposed to revise the writing at key points (including Table 1), and instead mention **infinite layouts**. Unfortunately, we never received a response from the reviewer. It is important to note that in response to other reviewers we introduced new sources of non-stationary elements that vary between the tasks in the sequence. Perhaps this is sufficient to keep the notion of **infinite tasks**.

### Minor issues

1. MEAL is not the **first JAX continual RL library**.

This is true, but we never claimed this. MEAL is a **benchmark**, and we explained the key differences.

2. **Neuron dormancy changes** are too small to compare methods.

We clarified that: (1) plasticity loss / neuron dormancy is not a core metric in MEAL, and (2) we only evaluated one baseline, so we don't whether this concern also applies to other methods.

3. MEAL **reimplements** a lot of functionality from Overcooked.

This is true and we were open about this from the beginning. We don't claim novelty of designing the environment from scratch. Why reinvent the wheel? Overcooked has been well-established in the MARL community. We focus on making a CL challenge in the MARL setting.

4. Algorithms trained on MEAL **perform too well**.

This is true in the Level 1 20-task sequence. However, we wish to highlight that performance is quick to collapse when using (1) single-output heads, (2) higher levels of difficuly, (3) more agents, and (4) longer sequences.

5. **Multi-head outputs** and **privileged information**.

We clarified that this is not the sole applicable architecture to use the benchmark. MEAL is agnostic to the model architecture, and the standard single-head approach without task-ID access is closer to a *true* CRL setting. We simply followed prior CRL benchmarks (Continual World, COOM), using the multi-head arch setup in our experiments, to create an easier setting. This allows for baseline methods to remain analyzable and do not collapse to near-zero performance, allowing us to meaningfully compare them.

6. Why do **harder tasks** cause more plasticity loss?

We clarified that we neither claimed nor empirically showed this. We only showed that plasticity loss increases with **more tasks**.

### Strengths
Reviewer gMyk praised us for having a good presentation and discussion, useful ablations and performance measures, and interesting additions and adjacent settings. The reviewer acknowledged the need for long task sequences in CRL.

### Resolution
**Reviewer gMyk never responded to our rebuttal.**

## Reviewer n4Rh

### Substantial concerns

1. Non-stationarity limited to layout changes

We added four new dynamic changes: (1) pot size variation, (2) soup timer variation, (3) sticky actions, and (4) slippery tiles. Each showed a difficulty increase, especially when combining all of them.

2. Only Overcooked

We explained that we intentionally chose to focus on a single domain and provide elaborate analysis, rather than scratch the surface of many environments. As proof that MEAL's CL pipeline is general, and can easily be extended to other envs, we incorporated **JaxNav** from JaxMARL as a proof-of-concept second environment. We experimented with a relatively simple setting across different number of agents and found the environment meaningful for CRL.

3. Scaling to more agents

We took it to the extremes and compared the performance and wall-clock times of 10, 25, and **100 agents**. We showed that while scaling is not linear, it is very much feasible to sensible extents.

**[Continued in the next response]**

---

> ### Author Response · Authors · 2025-11-29
> **Rebuttal Summary #2**
>
> ### Minor issues
>
> 1. Truly **continual RL**?
>
> The reviewer compared MEAL to CORA, COOM, Continual World, and Procgen-CL to argue that MEAL lacks variety and non-stationarity. We addressed each benchmark individually, explaining how MEAL’s formulation is fully aligned with standard continual RL practice and is not weaker or “less continual” than these benchmarks. We also explained why keeping observation and action spaces fixed is common and often desirable in CL. Finally, we added new factors of non-stationarity to further increase variety and complexity, as mentioned above.
>
> 2. Only **3 levels** and **infinite tasks**?
>
> We explained that scaling up difficulty levels is trivial and unbounded, so we didn't see a need to study this extensively, since no more interesting conclusions can be drawn from it. Regarding infinite tasks, the response was similar to Reviewer gmYk.
>
> 3. **Multi-head architectures** are inappropriate for *infinite* tasks.
>
> Again, effectively the same concern as raised by **Reviewer gMyk**, so our response was very similar.
>
> ### Strengths
> The reviewer praised us for an interesting and novel problem setting and for being the first to address the continual MARL paradigm. They found the paper to be well-written and organized, and praised the use of standard continual RL evaluation metrics.
>
> ### Resolution
> **Reviewer n4Rh never responded to our rebuttal.**
>
>
> ## Reviewer JNx1
>
> ### Substantial concerns
>
> 1. Limited task diversity.
>
> Similar response as to previous reviewers. (1) Elaborated on the differences between MEAL and other benchmarks. (2) Introduced JaxNav. (3) 4 new sources of non-stationarity. The reviewer appreciated these updates and indicated that the remaining concern was mainly about how MEAL is framed. We revised the paper accordingly: the main text now explicitly states MEAL’s intended scope, what its design targets, and what it consciously leaves out.
>
> ### Minor issues
>
> 1. An existing work has already studied CMARL, so **MEAL is not the first**.
>
> Clarified that we only claimed that MEAL is the first **benchmark** for CMARL.
>
> 2. Why no results for "***No Task ID***"?
>
> Clarified that the “*No Task ID*” ablation was already present in Figure 6.
>
> 3. Only **discrete action spaces**?
>
> Argued that this is not a limitation, since discrete RL is widely used in both real-world domains and recent RL benchmarks. The reviewer agreed and clarified that their concern was about positioning, not capability. In response, we revised the paper to more clearly articulate MEAL’s design scope, its intended use cases, and the types of CL challenges it is best suited to evaluate.
>
> 4. **Overcooked** not introduced.
>
> Although we already introduced overcooked at the end of Section 2, we added an additional short explanation at the beginning of Section 4.
>
> 5. **Acronyms** not explained.
>
> Agreed to fully write out the CL method names in the paper.
>
> 6. Why aren't there **per-task evaluation curves** of all baselines?
>
> Clarified that they are in the Appendix (Figure 17). Added a cross-reference and short summary in the main text.
>
> ### Strengths
>
> The Reviewer praised the timeliness of MEAL as a benchmark for CMARL, the value of its GPU-accelerated JAX implementation, insightful ablations, meaningful visualizations, the clear structure and organization of the paper, the strong empirical evaluation across CL metrics, and the breadth of implemented baselines. They also noted the thoughtful design of different evaluation settings (e.g., curriculum, repetition) and the usefulness of the PCG framework.
>
>
> ### Resolution
>
> We engaged in a thorough discussion with this reviewer, clarifying misunderstandings on both sides. Midway through, the reviewer raised their score to 6, and in their final comment stated that they are willing to further increase the score.

---

> ### Author Response · Authors · 2025-11-29
> **Rebuttal Summary #3**
>
> ## Reviewer tMtN
>
> ### Substantial concerns
>
> 1. MEAL is “***simply a level generator***”.
>
> We justified our focus on a single domain, and argued that MEAL provides far more than just layout generation, including the full CL pipeline with continual evaluation, N-agent support, a unified metric, difficulty scaling, partner adaptation, partial observability, etc. Since the reviewer requested to see at least one more environment, we added JaxNav to show that the framework is compatible with other environments. We made use of JaxNav's built-in features to generate layouts and showed that performance decreases with more agents.
>
> 2. Why doesn't the layout generator **force cooperation**?
>
> We acknowledged the utility of this proposition and added a feature to toggle the *forced_cooperation* mode. This mode enforces no inter-agent path connectivity and ensures that each agent cannot contribute to at least one segment of the cook–deliver cycle. We showed that under this setting, performance drops more rapidly with increasing difficulty levels.
>
> 3. You claim **100-task**, but we don't see it.
>
> This was a fair point. We included 100-task runs comparing EWC vs Online EWC across all three difficulty levels. This comparison demonstrates the importance of using long task sequences. While the performance was similar by 20 tasks, it diverged significantly by 100.
>
> 4. Why is the **network so small**?
>
> We compared our original MLP [128,128] with [256,256,256], which showed notable performance gains.
>
> ### Minor issues
>
> 1. A **GitHub repository** with a similar setting.
>
> This repository appears to be deleted/private.
>
> 2. Introduce **Overcooked**
>
> Same response as to the previous reviewer.
>
> 3. The **multi-head architecture** is mentioned too late
>
> We moved this to the beginning of Section 5.
>
> 4. Why so **few parallel envs**?
>
> An oversight by us. We updated the hyperparameter table to match the true values: 2048 envs, rollout 400, 1e8 steps/task.
>
>
> ### Strengths
> The reviewer praised the clarity of the paper, the value of MEAL’s JAX-based speedup for CRL where long task sequences make training expensive, and the extensive empirical analysis that gives a thorough picture of baseline behavior and future challenges.
>
>
> ### Resolution
> The reviewer raised their score to 6, without stating whether any of their concerns remained unaddressed. We were confused why the score remained only borderline given this assessment, but the reviewer did not provide any further clarification.
>
>
> Last, we would like to note that MEAL is a resubmission from **NeurIPS 2025 DB Track** (https://openreview.net/forum?id=DlBSkCQGAw), where it received **all positive scores and an acceptance recommendation from the AC**. However, this decision was overruled by the PCs due to a lack of physical space at the venue. We believe this context may be informative when making the final assessment.

---

### Meta-Review · Area_Chair_zY1J · 2025-12-29

**Summary:**

One main concern raised during the rebuttal was the precise relationship between the proposed benchmark and those used in prior work. This was partially addressed during the rebuttal, where the authors clarified that some of the papers cited by reviewers do not actually propose new benchmarks [JNx1]. However, other concerns here (e.g., about whether the proposed benchmark was "more general" [JNx1] and had sufficient levels [tMtN]) seem to have been somewhat unaddressed.

The second main concern was about the multi-head architecture used in the paper, which effectively leaks information about which tasks are different from one another. While the authors noted that this assumption is made in prior work, reviewers still requested that this detail be clearly stated [tMtN].

Overall, I (the AC) share the reviewer's concern about overlap with prior work. My understanding is that the benchmark proposed in this paper takes one task from a prior benchmark (Overcooked from JaxMARL) and augments it with additional "meals" so that it can be used for studying continual learning. As such, it seems like the main contribution of this paper isn't the benchmark but whatever scientific findings that benchmark enables. I think the paper _does_ have many of those interesting scientific findings. However, bringing them to the fore will require substantial revisions to the paper. Thus, I recommend that the paper be rejected and resubmitted to a future conference.

**Reviewer Concerns:**

(see below)

**Reviewer Scores:**

gMyk: 4 --> 6
* [/] Relationship with prior benchmarks (ReDo, Plasticine, Overcooked): Authors clarified that their paper proposes a benchmark, not an RL algorithm. They also discussed changing phrasing to clarify claims about the number of tasks
* [+] using different output heads for different tasks: Authors note that this is also done in prior benchmark papers, and without this the scores would be all 0, making comparisons difficult.

n4Rh: 2 --> 4
* [+] clarifying the notion of non-stationarity considered: authors acknowledged that the observation/action spaces are fixed, and extended the benchmark to add 4 other forms of non-stationarity
* [+] Use of procedural generation is not sufficient to claim infinite tasks: authors revised the wording to clarify this point.
* [+] primarily focuses on a multi-head output setting, instead of using a shared output head: Authors note that this is also done in prior benchmark papers, and without this the scores would be all 0, making comparisons difficult.

JNx1: 4 --> 4
* [+] works such as “Multi-Agent Continual Coordination via Progressive Task Contextualization” already explore continual multi-agent coordination: Authors argue that this paper is about an algorithm, rather than a benchmark. Reading the referenced paper, I agree with the authors, as that paper seems not to propose new environments but rather evaluates on tasks from prior work.
* [+] baseline results without task ID access: The original paper already includes these results.
* [/] limited task diversity, in comparison to other benchmarks (melting pot, continual world): Authors argue that these other tasks are not more "general purpose."
* [+] revise text to be accessible to new readers: Authors have promised to add this.
After back-and-forth discussion between the author and reviewer, it seems like the reviewer was likely to increase their score: "Given the authors' nuanced and detailed responses throughout this discussion, their willingness to incorporate feedback, and the concrete revisions made to the manuscript, I am willing to increase my score." However, the combative tone taken by the authors (e.g., "Please enlighten us on how these other toy-problem..."; "We are very curious where this allergy to discrete action spaces originates from") may have made the reviewer unsure about whether the final manuscript would provide a balanced presentation of prior work.

tMtN: 4 --> 6
* [-] relationship with overcooked jax: Authors agreed with this concern, but tried to argue that all benchmarks are "just a bundle of levels"
* relationship with jaxMARL
* [+] "In my opinion, while this benchmark can be used to assess different baselines and is useful, it should provide more than just a level generator for one environment.": Authors added a set of environments based on navigation.
* [+] "authors should include another level generator that forces the cooperation between the agents": Authors added an experiment to this effect.
* [/] "reader's understanding of the basic environment dynamics should not require them to look beyond the reviewed paper": Authors promised to add 2 sentences.
* [+] use of multi-headed architectures should be mentioned earlier: Authors will revise this.
* [+] "100 tasks", while only 20 used in the paper: Authors will clarify this by adding new experiments)
* [+] why is the number of parallel environments so small: authors fixed this typo.
The reviewer said they were raising their score to 6 during the rebuttal.

---

### Decision · Program_Chairs · 2026-01-26

Reject